# Neural Delay Differential Equations

**Qunxi Zhu, Yao Guo & Wei Lin** *
School of Mathematical Science, Research Institute of Intelligent Complex Systems and ISTBI
State Key Laboratory of Medical Neurobiology and MOE Frontiers Center for Brain Science
Fudan University
Shanghai 200433, China
{qxzhu16,yguo,wlin}@fudan.edu.cn

## Abstract

Neural Ordinary Differential Equations (NODEs), a framework of continuous-depth neural networks, have been widely applied, showing exceptional efficacy in coping with some representative datasets. Recently, an augmented framework has been successfully developed for conquering some limitations emergent in application of the original framework. Here we propose a new class of continuous-depth neural networks with delay, named as Neural Delay Differential Equations (ND-DEs), and, for computing the corresponding gradients, we use the adjoint sensitivity method to obtain the delayed dynamics of the adjoint. Since the differential equations with delays are usually seen as dynamical systems of infinite dimension possessing more fruitful dynamics, the NDDEs, compared to the NODEs, own a stronger capacity of nonlinear representations. Indeed, we analytically validate that the NDDEs are of universal approximators, and further articulate an extension of the NDDEs, where the initial function of the NDDEs is supposed to satisfy ODEs. More importantly, we use several illustrative examples to demonstrate the outstanding capacities of the NDDEs and the NDDEs with ODEs' initial value. Specifically, (1) we successfully model the delayed dynamics where the trajectories in the lower-dimensional phase space could be mutually intersected, while the traditional NODEs without any argumentation are not directly applicable for such modeling, and (2) we achieve lower loss and higher accuracy not only for the data produced synthetically by complex models but also for the real-world image datasets, i.e., CIFAR10, MNIST, and SVHN. Our results on the NDDEs reveal that appropriately articulating the elements of dynamical systems into the network design is truly beneficial to promoting the network performance.

## 1 Introduction

A series of recent works have revealed a close connection between neural networks and dynamical systems (E, 2017; Li et al., 2017; Haber & Ruthotto, 2017; Chang et al., 2017; Li & Hao, 2018; Lu et al., 2018; E et al., 2019; Chang et al., 2019; Ruthotto & Haber, 2019; Zhang et al., 2019a; Pathak et al., 2018; Fang et al., 2018; Zhu et al., 2019; Tang et al., 2020). On one hand, the deep neural networks can be used to solve the ordinary/partial differential equations that cannot be easily computed using the traditional algorithms. On the other hand, the elements of the dynamical systems can be useful for establishing novel and efficient frameworks of neural networks. Typical examples include the Neural Ordinary Differential Equations (NODEs), where the infinitesimal time of ordinary differential equations is regarded as the "depth" of a considered neural network (Chen et al., 2018).

Though the advantages of the NODEs were demonstrated through modeling continuous-time datasets and continuous normalizing flows with constant memory cost (Chen et al., 2018), the limited capability of representation for some functions were also studied (Dupont et al., 2019). Indeed, the NODEs cannot be directly used to describe the dynamical systems where the trajectories in the lower-dimensional phase space are mutually intersected. Also, the NODEs cannot model only a few variables from some physical or/and physiological systems where the effect of time delay is

---

*http://homepage.fudan.edu.cn/weilin, To whom correspondence should be addressed: Q.Z., Y.G., and W.L.

inevitably present. From a view point of dynamical systems theory, all these are attributed to the characteristic of finite-dimension for the NODEs.

In this article, we propose a novel framework of continuous-depth neural networks with delay, named as Neural Delay Differential Equations (NDDEs). We apply the adjoint sensitivity method to compute the corresponding gradients, where the obtained adjoint systems are also in a form of delay differential equations. The main virtues of the NDDEs include:

- feasible and computable algorithms for computing the gradients of the loss function based on the adjoint systems,

- representation capability of the vector fields which allow the intersection of the trajectories in the lower-dimensional phase space, and

- accurate reconstruction of the complex dynamical systems with effects of time delays based on the observed time-series data.

## 2 RELATED WORKS

**NODEs** Inspired by the residual neural networks (He et al., 2016) and the other analogous frameworks, the NODEs were established, which can be represented by multiple residual blocks as

$$\boldsymbol{h}_{t+1} = \boldsymbol{h}_t + \boldsymbol{f}(\boldsymbol{h}_t, \boldsymbol{w}_t),$$

where $\boldsymbol{h}_t$ is the hidden state of the $t$-th layer, $\boldsymbol{f}(\boldsymbol{h}_t; \boldsymbol{w}_t)$ is a differential function preserving the dimension of $\boldsymbol{h}_t$, and $\boldsymbol{w}_t$ is the parameter pending for learning. The evolution of $\boldsymbol{h}_t$ can be viewed as the special case of the following equation

$$\boldsymbol{h}_{t+\Delta t} = \boldsymbol{h}_t + \Delta t \cdot \boldsymbol{f}(\boldsymbol{h}_t, \boldsymbol{w}_t)$$

with $\Delta t = 1$. As suggested in (Chen et al., 2018), all the parameters $\boldsymbol{w}_t$ are unified into $\boldsymbol{w}$ for achieving parameter efficiency of the NODEs. This unified operation was also employed in the other neural networks, such as the recurrent neural networks (RNNs) (Rumelhart et al., 1986; Elman, 1990) and the ALBERT (Lan et al., 2019). Letting $\Delta t \to 0$ and using the unified parameter $\boldsymbol{w}$ instead of $\boldsymbol{w}_t$, we obtain the continuous evolution of the hidden state $\boldsymbol{h}_t$ as

$$\lim_{\Delta t \to 0} \frac{\boldsymbol{h}_{t+\Delta t} - \boldsymbol{h}_t}{\Delta t} = \frac{d\boldsymbol{h}_t}{dt} = \boldsymbol{f}(\boldsymbol{h}_t, t; \boldsymbol{w}),$$

which is in the form of ordinary differential equations. Actually, the NODEs can act as a feature extraction, mapping an input to a point in the feature space by computing the forward path of a NODE as:

$$\boldsymbol{h}(T) = \boldsymbol{h}(0) + \int_0^T \boldsymbol{f}(\boldsymbol{h}_t, t; \boldsymbol{w})dt, \ \ \boldsymbol{h}(0) = \text{input},$$

where $\boldsymbol{h}(0) = \text{input}$ is the original data point or its transformation, and $T$ is the integration time (assuming that the system starts at $t = 0$).

Under a predefined loss function $L(\boldsymbol{h}(T))$, (Chen et al., 2018) employed the adjoint sensitivity method to compute the memory-efficient gradients of the parameters along with the ODE solvers. More precisely, they defined the adjoint variable, $\boldsymbol{\lambda}(t) = \frac{\partial L(\boldsymbol{h}(T))}{\partial \boldsymbol{h}(t)}$, whose dynamics is another ODE, i.e.,

$$\frac{d\boldsymbol{\lambda}(t)}{dt} = -\boldsymbol{\lambda}(t)^\top \frac{\partial f(\boldsymbol{h}_t, t; \boldsymbol{w})}{\partial \boldsymbol{h}_t}. \tag{1}$$

The gradients are computed by an integral as:

$$\frac{dL}{d\boldsymbol{w}} = \int_T^0 -\boldsymbol{\lambda}(t)^\top \frac{\partial f(\boldsymbol{h}_t, t; \boldsymbol{w})}{\partial \boldsymbol{w}}dt. \tag{2}$$

(Chen et al., 2018) calculated the gradients by calling an ODE solver with extended ODEs (i.e., concatenating the original state, the adjoint, and the other partial derivatives for the parameters at each time point into a single vector). Notably, for the regression task of the time series, the loss function probably depend on the state at multiple observational times, such as the form of

$L(h(t_0), h(t_1), ..., h(t_n))$. Under such a case, we must update the adjoint state instantly by adding the partial derivative of the loss at each observational time point.

As emphasized in (Dupont et al., 2019), the flow of the NODEs cannot represent some functions omnipresently emergent in applications. Typical examples include the following two-valued function with one argument: $g_{\text{1-D}}(1) = -1$ and $g_{\text{1-D}}(-1) = 1$. Our framework desires to conquer the representation limitation observed in applying the NODEs.

**Optimal control** As mentioned above, a closed connection between deep neural networks and dynamical systems have been emphasized in the literature and, correspondingly, theories, methods and tools of dynamical systems have been employed, e.g. the theory of optimal control (E, 2017; Li et al., 2017; Haber & Ruthotto, 2017; Chang et al., 2017; Li & Hao, 2018; E et al., 2019; Chang et al., 2019; Ruthotto & Haber, 2019; Zhang et al., 2019a). Generally, we model a typical task using a deep neural network and then train the network parameters such that the given loss function can be reduced by some learning algorithm. In fact, training a network can be seen as solving an optimal control problem on difference or differential equations (E et al., 2019). The parameters act as a controller with a goal of finding an optimal control to minimize/maximize some objective function. Clearly, the framework of the NODEs can be formulated as a typical problem of optimal control on ODEs. Additionally, the framework of NODEs has been generalized to the other dynamical systems, such as the Partial Differential Equations (PDEs) (Han et al., 2018; Long et al., 2018; 2019; Ruthotto & Haber, 2019; Sun et al., 2020) and the Stochastic Differential Equations (SDEs) (Lu et al., 2018; Sun et al., 2018; Liu et al., 2019), where the theory of optimal control has been completely established. It is worthwhile to mention that the optimal control theory is tightly connected with and benefits from the method of the classical calculus of variations (Liberzon, 2011). We also will transform our framework into an optimal control problem, and finally solve it using the method of the calculus of variations.

## 3 THE FRAMEWORK OF NDDES

### 3.1 FORMULATION OF NDDES

In this section, we establish a framework of continuous-depth neural networks. To this end, we first introduce the concept of delay deferential equations (DDEs). The DDEs are always written in a form where the derivative of a given variable at time $t$ is affected not only by the current state of this variable but also the states at some previous time instants or time durations (Erneux, 2009). Such kind of delayed dynamics play an important role in description of the complex phenomena emergent in many real-world systems, such as physical, chemical, ecological, and

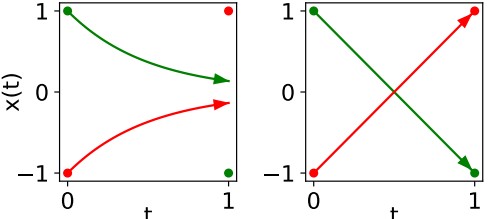

Figure 2: (Right) Two continuous trajectories generated by the DDEs are intersected, mapping -1 (resp., 1) to 1 (resp., -1), while (Left) the ODEs cannot represent such mapping.

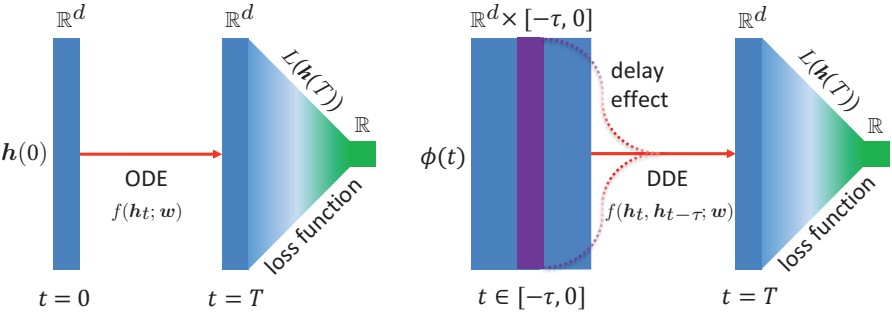

Figure 1: Sketchy diagrams of the NODEs and the NDDES, respectively, with the initial value $h(0)$ and the initial function $\phi(t)$. The NODEs and the NDDEs act as the feature extractors, and the following layer processes the features with a predefined loss function.

physiological systems. In this article, we consider a system of DDE with a single time delay:

$$\begin{cases} \frac{d\boldsymbol{h}_t}{dt} = f(\boldsymbol{h}_t, \boldsymbol{h}_{t-\tau}, t; \boldsymbol{w}), & t >= 0, \\ \boldsymbol{h}(t) = \phi(t), & t <= 0, \end{cases}$$

where the positive constant $\tau$ is the time delay and $\boldsymbol{h}(t) = \phi(t)$ is the *initial function* before the time $t = 0$. Clearly, in the initial problem of ODEs, we only need to initialize the state of the variable at $t = 0$ while we initialize the DDEs using a continuous function. Here, to highlight the difference between ODEs and DDEs, we provide a simple example:

$$\begin{cases} \frac{dx_t}{dt} = -2x_{t-\tau}, & t >= 0, \\ x(t) = x_0, & t <= 0. \end{cases}$$

where $\tau = 1$ (with time delay) or $\tau = 0$ (without time delay). As shown in Figure 2, the DDE flow can map -1 to 1 and 1 to -1; nevertheless, this cannot be made for the ODE whose trajectories are not intersected with each other in the $t$-$x$ space in Figure 2.

## 3.2 ADJOINT METHOD FOR NDDES

Assume that the forward pass of DDEs is complete. Then, we need to compute the gradients in a reverse-mode differentiation by using the adjoint sensitivity method (Chen et al., 2018; Pontryagin et al., 1962). We consider an augmented variable, named as *adjoint* and defined as

$$\boldsymbol{\lambda}(t) = \frac{\partial L(\boldsymbol{h}(T))}{\partial \boldsymbol{h}(t)}, \tag{3}$$

where $L(\cdot)$ is the loss function pending for optimization. Notably, the resulted system for the adjoint is in a form of DDE as well.

**Theorem 1** *(Adjoint method for NDDEs). Consider the loss function $L(\cdot)$. Then, the dynamics of adjoint can be written as*

$$\begin{cases} \dfrac{d\boldsymbol{\lambda}(t)}{dt} = -\boldsymbol{\lambda}(t)^\top \dfrac{\partial f(\boldsymbol{h}_t, \boldsymbol{h}_{t-\tau}, t; \boldsymbol{w})}{\partial \boldsymbol{h}_t} - \boldsymbol{\lambda}(t+\tau)^\top \dfrac{\partial f(\boldsymbol{h}_{t+\tau}, \boldsymbol{h}_t, t; \boldsymbol{w})}{\partial \boldsymbol{h}_t} \chi_{[0,T-\tau]}(t), \ t <= T \\ \boldsymbol{\lambda}(T) = \dfrac{\partial L(\boldsymbol{h}(T))}{\partial \boldsymbol{h}(T)}, \end{cases}$$

$$\tag{4}$$

*where $\chi_{[0,T-\tau]}(\cdot)$ is a typical characteristic function.*

We provide two ways to prove Theorem 1, which are, respectively, shown in Appendix. Using $\boldsymbol{h}(t)$ and $\boldsymbol{\lambda}(t)$, we compute the gradients with respect to the parameters $\boldsymbol{w}$ as:

$$\frac{dL}{d\boldsymbol{w}} = \int_T^0 -\boldsymbol{\lambda}(t)^\top \frac{\partial f(\boldsymbol{h}_t, \boldsymbol{h}_{t-\tau}, t; \boldsymbol{w})}{\partial \boldsymbol{w}} dt. \tag{5}$$

Clearly, when the delay $\tau$ approaches zero, the adjoint dynamics degenerate as the conventional case of the NODEs (Chen et al., 2018).

We solve the forward pass of $\boldsymbol{h}$ and backward pass for $\boldsymbol{h}$, $\boldsymbol{\lambda}$ and $\frac{dL}{d\boldsymbol{w}}$ by a piece-wise ODE solver, which is shown in Algorithm 1. For simplicity, we denote by $f(t)$ and $g(t)$ the vector filed of $\boldsymbol{h}$ and $\boldsymbol{\lambda}$, respectively. Moreover, in this paper, we only consider the initial function $\phi(t)$ as a constant function, i.e., $\phi(t) = \boldsymbol{h}_0$. Assume that $T = n \cdot \tau$ and denote $f_k(t) = f(k \cdot \tau + t)$, $g_k(t) = g(k \cdot \tau + t)$ and $\boldsymbol{\lambda}_k(t) = \boldsymbol{\lambda}(k \cdot \tau + t)$.

In the traditional framework of the NODEs, we can calculate the gradients of the loss function and recompute the hidden states by solving another augmented ODEs in a reversal time duration. However, to achieve the reverse-mode of the NDDEs in Algorithm 1, we need to store the checkpoints of the forward hidden states $h(i \cdot \tau)$ for $i = 0, 1, ..., n$, which, together with the adjoint $\boldsymbol{\lambda}(t)$, can help us to recompute $h(t)$ backwards in every time periods. The main idea of the Algorithm 1 is to convert the DDEs as a piece-wise ODEs such that one can naturally employ the framework of the NODEs to solve it. The complexity of Algorithm 1 is analyzed in Appendix.

---

**Algorithm 1** Piece-wise reverse-mode derivative of an DDE initial function problem

---

**Input:** dynamics parameters $\boldsymbol{w}$, time delay $\tau$, start time 0, stop time $T = n \cdot \tau$, final state $\boldsymbol{h}(T)$, loss gradient $\partial L / \partial \boldsymbol{h}(T)$

$\quad \frac{\partial L}{\partial \boldsymbol{w}} = 0_{|\boldsymbol{w}|}$

$\quad$ **for** $i$ **in range**$(n-1, -1, -1)$:

$\quad\quad s_0 = [\boldsymbol{h}(T), \boldsymbol{h}(T-\tau), ..., \boldsymbol{h}(\tau), \frac{\partial L}{\partial \boldsymbol{h}(T)}, ..., \frac{\partial L}{\partial \boldsymbol{h}((i+1)\cdot\tau)}, \frac{\partial L}{\partial \boldsymbol{w}}]$

$\quad\quad$ **def** aug_dynamics$([\boldsymbol{h}_{n-1}(t),...,\boldsymbol{h}_0(t), \boldsymbol{\lambda}_{n-1}(t),..,\boldsymbol{\lambda}_i(t), .], t, \boldsymbol{w})$:

$\quad\quad\quad$ **return** $[f_{n-1}(t), f_{n-2}(t), ..., f_0(t), g_{n-1}(t), ..., g_i(t), -\boldsymbol{\lambda}_i(t)^\top \frac{\partial f_i(t)}{\partial \boldsymbol{w}}]$

$\quad\quad [\frac{\partial L}{\partial \boldsymbol{h}(i\cdot\tau)}, \frac{\partial L}{\partial \boldsymbol{w}}]$=ODESolve$(s_0,$ aug_dynamics$, \tau, 0, \boldsymbol{w})$

**return** $\frac{\partial L}{\partial \boldsymbol{h}(0)}, \frac{\partial L}{\partial \boldsymbol{w}}$

---

# 4 ILLUSTRATIVE EXPERIMENTS

## 4.1 EXPERIMENTS ON SYNTHETIC DATASETS

Here, we use some synthetic datasets produced by typical examples to compare the performance of the NODES and the NDDEs. In (Dupont et al., 2019), it is proved that the NODEs cannot represent the function $g : \mathbb{R}^d \to \mathbb{R}$, defined by

$$g(\boldsymbol{x}) = \begin{cases} 1, & \text{if } \|\boldsymbol{x}\| \le r_1, \\ -1, & \text{if } r_2 \le \|\boldsymbol{x}\| \le r_3, \end{cases}$$

where $0 < r_1 < r_2 < r_3$ and $\| \cdot \|$ is the Euclidean norm. The following proposition show that the NDDEs have stronger capability of representation.

**Proposition 1** *The NDDEs can represent the function $g(\boldsymbol{x})$ specified above.*

To validate this proposition, we construct a special form of the NDDEs by

$$\begin{cases} \frac{d\boldsymbol{h}_i(t)}{dt} = \|\boldsymbol{h}_{t-\tau}\| - r, & t >= 0 \text{ and } i = 1, \\ \frac{d\boldsymbol{h}_i(t)}{dt} = 0, & t >= 0 \text{ and } i = 2, ..., d, \\ \boldsymbol{h}(t) = \boldsymbol{x}, & t <= 0. \end{cases} \tag{6}$$

where $r := (r_1 + r_2)/2$ is a constant and the final time point $T$ is supposed to be equal to the time delay $\tau$ with some sufficient large value. Under such configurations, we can linearly separate the two clusters by some hyper plane.

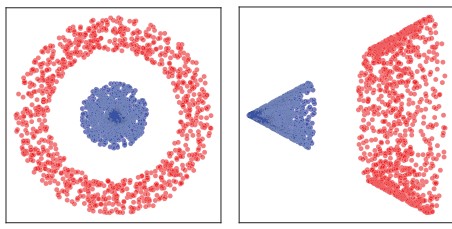

Figure 3: (Left) The data at the time $t = 0$ and (Right) the transformed data at a sufficient large final time $T$ of the DDEs (6). Here, the transformed data are linearly separable.

In Fig. 3, we, respectively, show the original data of $g(\boldsymbol{x})$ for the dimension $d = 2$ and the transformed data by the DDEs (6) with the parameters $r_1 = 1$, $r_2 = 2$, $r_3 = 3$, $T = 10$, and $\tau = 10$. Clearly, the transformed data by the DDEs are linearly separable. We also train the NDDEs to represent the $g(\boldsymbol{x})$ for $d = 2$, whose evolutions during the training procedure are, respectively, shown in Fig. 4. This figure also includes the evolution of the NODEs. Notably, while the NODEs is struggled to break apart the annulus, the NDDEs easily separate them. The training losses and the flows of the NODEs and the NDDEs are depicted, respectively, in Fig. 5. Particularly, the NDDEs achieve lower losses with the faster speed and directly separate the two clusters in the original 2-D space; however, the NODEs achieve it only by increasing the dimension of the data and separating them in a higher-dimensional space.

In general, we have the following theoretical result for the NDDEs, whose proof is provided in Appendix.

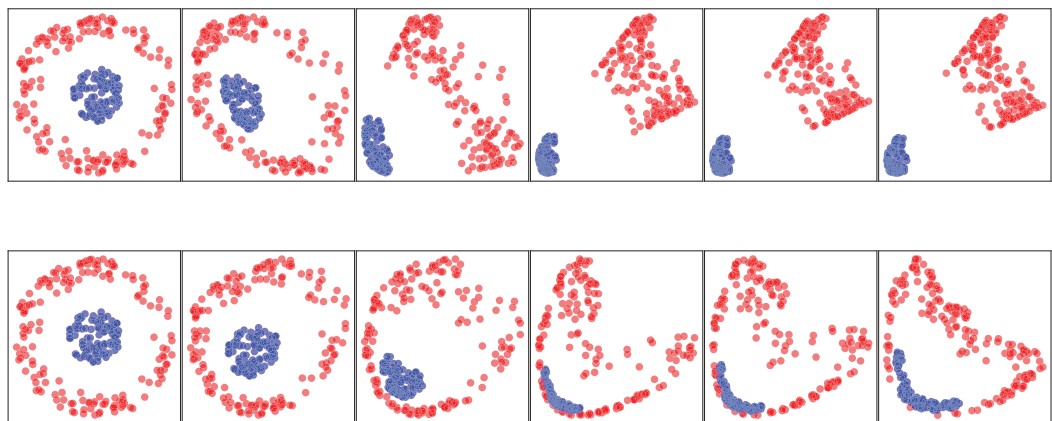

Figure 4: Evolutions of the NDDEs (top) and the NODEs (bottom) in the feature space during the training procedure. Here, the evolution of the NODEs is directly produced by the code provided in (Dupont et al., 2019).

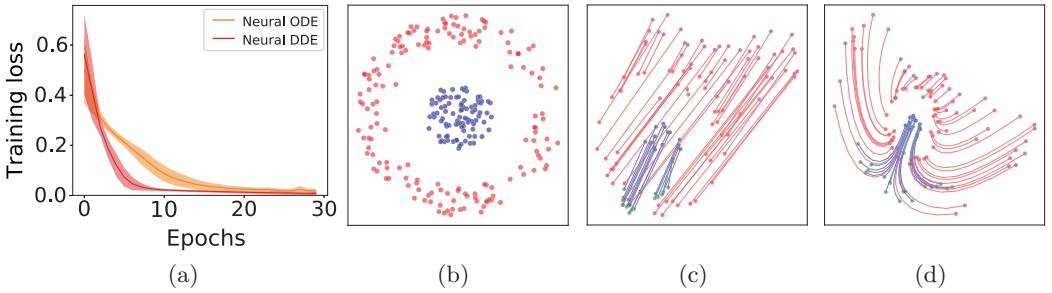

Figure 5: Presented are the training losses (a) of the NODEs and the NDDEs on fitting the function $g(x)$ for $d = 2$. Also presented are the flows, from the data at the initial time point (b), of the NODEs (d) and the NDDEs (c) after training. The flow of the NODEs is generated by the code provided in (Dupont et al., 2019).

**Theorem 2** *(Universal approximating capability of the NDDEs). For any given continuous function $F : \mathbb{R}^n \to \mathbb{R}^n$, if one can construct a neural network for approximating the map $G(\boldsymbol{x}) = \frac{1}{T}[F(\boldsymbol{x}) - \boldsymbol{x}]$, then there exists an NDDE of $n$-dimension that can model the map $\boldsymbol{x} \mapsto F(\boldsymbol{x})$, that is, $\boldsymbol{h}(T) \approx F(\boldsymbol{x})$ with the initial function $\phi(t) = \boldsymbol{x}$ for $t \leq 0$.*

Additionally, NDDEs are suitable for fitting the time series with the delay effect in the original systems, which cannot be easily achieved by using the NODEs. To illustrate this, we use a model of 2-D DDEs, written as $\dot{\boldsymbol{x}} = \boldsymbol{A} \tanh(\boldsymbol{x}(t) + \boldsymbol{x}(t - \tau))$ with $\boldsymbol{x}(t) = \boldsymbol{x}_0$ for $t < 0$. Given the time series generated by the system, we use the NDDEs and the NODEs to fit it, respectively. Figure 6 shows that the NNDEs approach a much lower loss, compared to the NODEs. More interestingly, the NODEs prefer to fit the dimension of $\boldsymbol{x}_2$ and its loss always sustains at a larger value, e.g. 0.25 in Figure 6. The main reason is that two different trajectories generated by the autonomous ODEs cannot intersect with each other in the phase space, due to the uniqueness of the solution of ODEs.

We perform experiments on another two classical DDEs, i.e., the population dynamics and the Mackey-Glass system (Erneux, 2009). Specifically, the equation of the dimensionless population dynamics is $\dot{x} = rx(t)(1 - x(t - \tau))$, where $x(t)$ is the ratio of the population to the carrying capacity of the population, $r$ is the growth rate and $\tau$ is the time delay. The Mackey-Glass system is written as $\dot{x} = \beta \frac{x(t-\tau)}{1+x^n(t-\tau)} - \gamma x(t)$, where $x(t)$ is the number of the blood cells, $\beta, n, \tau, \gamma$ are the parameters of biological significance. The NODEs and the NDDEs are tested on these two dynamics. As shown in Figure 7, a very low training loss is achieved for the NDDEs while the the

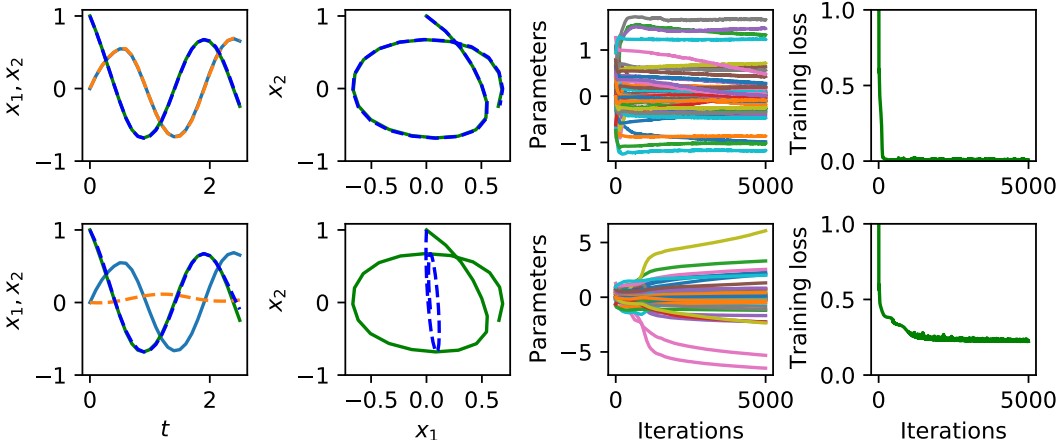

Figure 6: Comparison of the NDDEs (top) versus the NODEs (bottom) in the fitting a 2-D time series with the delay effect in the original system. From the left to the right, the true and the fitted time series, the true and the fitted trajectories in the phase spaces, the dynamics of the parameters in the neural networks, and the dynamics of the losses during the training processes.

loss of the NODEs does not go down afterwards, always sustaining at some larger value. As for the predicting durations, the NDDEs thereby achieve a better performance than the NODEs. The details of the training could be found in Appendix.

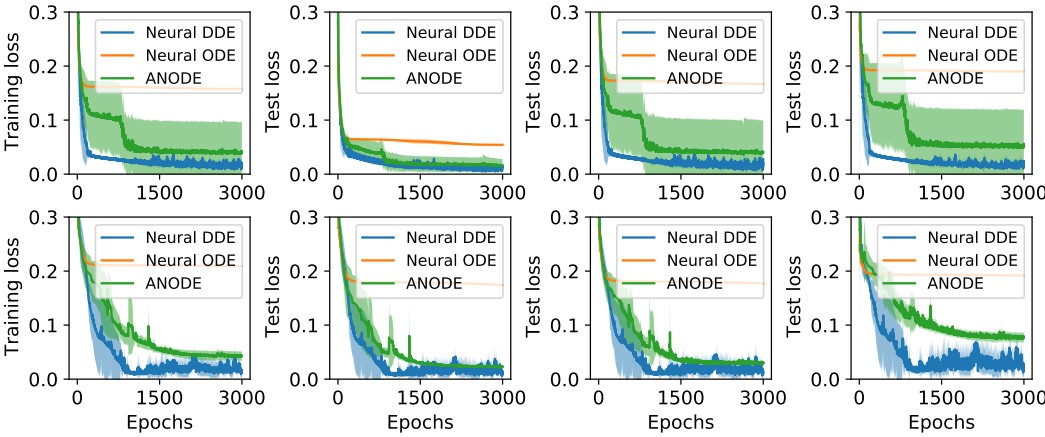

Figure 7: The training losses and the test losses of the population dynamics (top) and the Mackey-Glass system (bottom) by using the NDDEs, the NODEs and the ANODEs (where the augmented dimension equals to 1). The first column in the figures is the training losses. The figures from the second column to the fourth column present the test losses over the time intervals of the length $\tau$, $2\tau$, $5\tau$. The delays for the two DDEs are both designed as 1, respectively. The other parameters are set as $r = 1.8$, $\beta = 4.0$, $n = 9.65$, and $\gamma = 2$.

## 4.2 EXPERIMENTS ON IMAGE DATASETS

For the image datasets, we not only use the NODEs and the NDDEs but also an extension of the NDDEs, called the NODE+NDDE, which treats the initial function as an ODE. In our experiments, such a model exhibits the best performance, revealing strong capabilities in modelling and feature representations. Moreover, inspired by the idea of the augmented NODEs (Dupont et al., 2019), we extend the NDDEs and the NODE+NDDE to the A+NDDE and the A+NODE+NDDE, respectively. Precisely, for the augmented models, we augment the original image space to a higher-dimensional

space, i.e., $\mathbb{R}^{c \times h \times w} \to \mathbb{R}^{(c+p) \times h \times w}$, where $c$, $h$, $w$, and $p$ are, respectively, the number of channels, height, width of the image, and the augmented dimension. With such configurations of the same augmented dimension and approximately the same number of the model parameters, comparison studies on the image datasets using different models are reasonable. For the NODEs, we model the vector filed $f(\boldsymbol{h}(t))$ as the convolutional architectures together with a slightly different hyperparameter setups in (Dupont et al., 2019). The initial hidden state is set as $\boldsymbol{h}(0) \in \mathbb{R}^{c \times h \times w}$ with respect to an image. For the NDDEs, we design the vector filed as $f(\text{concat}(\boldsymbol{h}(t), \boldsymbol{h}(t - \tau)))$, mapping the space from $\mathbb{R}^{2c \times h \times w}$ to $\mathbb{R}^{c \times h \times w}$, where $\text{concat}(\cdot, \cdot)$ executes the concatenation of two tensors on the channel dimension. Its initial function is designed as a constant function, i.e., $h(t)$ is identical to the input/image for $t < 0$. For the NODE+NDDE, we model the initial function as an ODE, which follows the same model structure of the NODEs. For the augmented models, the augmented dimension is chosen from the set $\{1, 2, 4\}$. Moreover, the training details could be found in Appendix, including the training setting and the number of function evaluations for each model on the image datasets.

The training processes on MNIST, CIFAR10, and SVHN are shown in Fig. 8. Overall, the NDDEs and its extensions have their training losses decreasing faster than the NODEs/ANODEs which achieve lower training and test loss. Also the test accuracies are much higher than the that of the NODEs/ANODEs (refer to Tab. 1). Naturally, the better performance of the NDDEs is attributed to the integration of the information not only on the hidden states at the current time $t$ but at the previous time $t - \tau$. This kind of framework is akin to the key idea proposed in (Huang et al., 2017), where the information is processed on many hidden states. Here, we run all the experiments for 5 times independently.

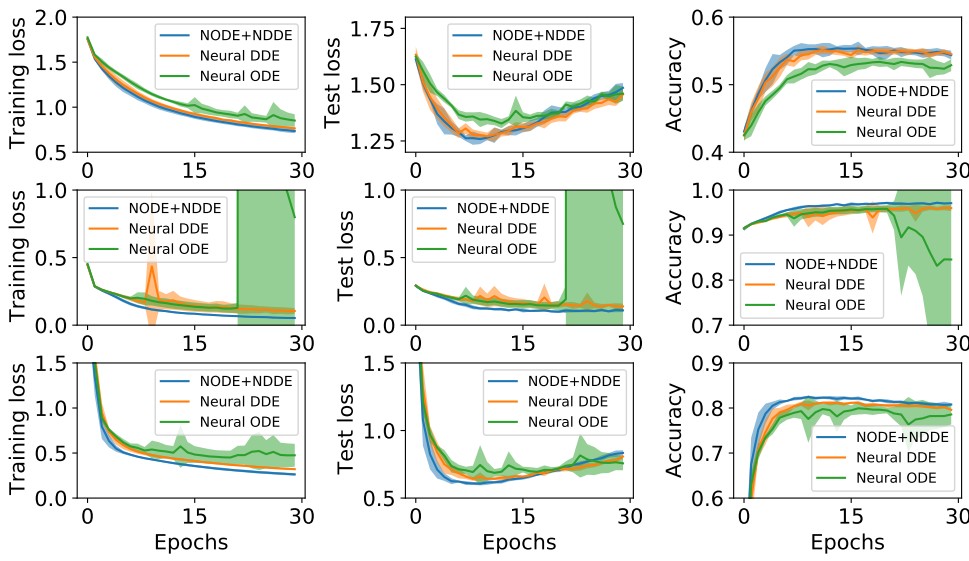

Figure 8: The training loss (left column), the test loss (middle column), and the accuracy (right column) over 5 realizations for the three image sets, i.e., CIFAR10 (top row), MNIST (middle row), and SVHN (bottom row).

## 5  DISCUSSION

In this section, we present the limitations of the NDDEs and further suggest several potential directions for future works. We add the delay effect to the NDDEs, which renders the model absolutely irreversible. Algorithm 1 thus requires a storage of the checkpoints of the hidden state $\boldsymbol{h}(t)$ at every instant of the multiple of $\tau$. Actually, solving the DDEs is transformed to solving the ODEs of an increasingly high dimension with respect to the ratio of the final time $T$ and the time delay $\tau$. This definitely indicates a high computational cost. To further apply and improve the framework of the NDDEs, a few potential directions for future works are suggested, including:

|           | CIFAR10 | MNIST | SVHN |
|-----------|---------|-------|------|
| NODE | $53.92\% \pm 0.67$ | $96.21\% \pm 0.66$ | $80.66\% \pm 0.56$ |
| NDDE | $55.69\% \pm 0.39$ | $96.22\% \pm 0.55$ | $81.49\% \pm 0.09$ |
| NODE+NDDE | $\mathbf{55.89\% \pm 0.71}$ | $\mathbf{97.26\% \pm 0.22}$ | $\mathbf{82.60\% \pm 0.22}$ |
| A1+NIDE | $56.14\% \pm 0.48$ | $97.89\% \pm 0.14$ | $81.17\% \pm 0.29$ |
| A1+NDDE | $56.83\% \pm 0.60$ | $97.83\% \pm 0.07$ | $82.46\% \pm 0.28$ |
| A1+NODE+NDDE | $\mathbf{57.31\% \pm 0.61}$ | $\mathbf{98.16\% \pm 0.07}$ | $\mathbf{83.02\% \pm 0.37}$ |
| A2+NODE | $57.27\% \pm 0.46$ | $98.25\% \pm 0.08$ | $81.73\% \pm 0.92$ |
| A2+NDDE | $58.13\% \pm 0.32$ | $98.22\% \pm 0.04$ | $82.43\% \pm 0.26$ |
| A2+NODE+NDDE | $\mathbf{58.40\% \pm 0.31}$ | $\mathbf{98.26\% \pm 0.06}$ | $\mathbf{83.73\% \pm 0.72}$ |
| A4+NODE | $58.93\% \pm 0.33$ | $98.33\% \pm 0.12$ | $82.72\% \pm 0.60$ |
| A4+NDDE | $59.35\% \pm 0.48$ | $98.31\% \pm 0.03$ | $82.87\% \pm 0.55$ |
| A4+NODE+NDDE | $\mathbf{59.94\% \pm 0.66}$ | $\mathbf{98.52\% \pm 0.11}$ | $\mathbf{83.62\% \pm 0.51}$ |

Table 1: The test accuracies with their standard deviations over 5 realizations on the three image datasets. In the first column, $p$ (=1, 2, or 4) in A$p$ means the number of the channels of zeros into the input image during the augmentation of the image space $\mathbb{R}^{c \times h \times w} \to \mathbb{R}^{(c+p) \times h \times w}$ (Dupont et al., 2019). For each model, the initial (resp. final) time is set as 0 (resp. 1), and the delays of the NDDEs and its extensions are all set as 1, simply equal to the final time.

**Applications to more real-world datasets.** In the real-world systems such as physical, chemical, biological, and ecological systems, the delay effects are inevitably omnipresent, truly affecting the dynamics of the produced time-series (Bocharov & Rihan, 2000; Kajiwara et al., 2012). The NDDEs are undoubtedly suitable for realizing model-free and accurate prediction (Quaglino et al., 2019). Additionally, since the NODEs have been generalized to the areas of Computer Vision and Natural Language Processing He et al. (2019); Yang et al. (2019); Liu et al. (2020), the framework of the NDDEs probably can be applied to the analogous areas, where the delay effects should be ubiquitous in those streaming data.

**Extension of the NDDEs.** A single constant time-delay in the NDDEs can be further generalized to the case of multiple or/and distributed time delays (Shampine & Thompson, 2001). As such, the model is likely to have much stronger capability to extract the feature, because the model leverages the information at different time points to make the decision in time. All these extensions could be potentially suitable for some complex tasks. However, such complex model may require tremendously huge computational cost.

**Time-dependent controllers.** From a viewpoint of control theory, the parameters in the NODEs/NDDEs could be regarded as time-independent controllers, viz. constant controllers. A natural generalization way is to model the parameters as time-dependent controllers. In fact, such controllers were proposed in (Zhang et al., 2019b), where the parameters $\boldsymbol{w}(t)$ were treated as another learnable ODE $\dot{\boldsymbol{w}} = q(\boldsymbol{w}(t), \boldsymbol{p})$, $q(\cdot, \cdot)$ is a different neural network, and the parameters $\boldsymbol{p}$ and the initial state $\boldsymbol{w}(0)$ are pending for optimization. Also, the idea of using a neural network to generate the other one was initially conceived in some earlier works including the study of the hypernetworks (Ha et al., 2016).

## 6 CONCLUSION

In this article, we establish the framework of NDDEs, whose vector fields are determined mainly by the states at the previous time. We employ the adjoint sensitivity method to compute the gradients of the loss function. The obtained adjoint dynamics backward follow another DDEs coupled with the forward hidden states. We show that the NDDEs can represent some typical functions that cannot be represented by the original framework of NODEs. Moreover, we have validated analytically that the NDDEs possess universal approximating capability. We also demonstrate the exceptional efficacy of the proposed framework by using the synthetic data or real-world datasets. All these reveal that integrating the elements of dynamical systems to the architecture of neural networks could be potentially beneficial to the promotion of network performance.

AUTHOR CONTRIBUTIONS

WL conceived the idea, QZ, YG & WL performed the research, QZ & YG performed the mathematical arguments, QZ analyzed the data, and QZ & WL wrote the paper.

ACKNOWLEDGMENTS

WL was supported by the National Key RD Program of China (Grant no. 2018YFC0116600), the National Natural Science Foundation of China (Grant no. 11925103), and by the STCSM (Grant No. 18DZ1201000, 19511101404, 19511132000).

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

## A  APPENDIX

### A.1  THE FIRST PROOF OF THEOREM 1

Here, we present a direct proof of the adjoint method for the NDDEs. For the neat of the proof, the following notations are slightly different from those in the main text.

Let $x(t)$ obey the DDE written as

$$\begin{cases} \dot{x}(t) = f(x(t), y(t), \theta(t)), \ y(t) = x(t - \tau), \ t \in [0, T], \\ x(t) = x_0, \ t \in [-\tau, 0]. \end{cases} \tag{7}$$

whose adjoint state is defined as

$$\lambda(t) := \frac{\partial L}{\partial x(t)}, \tag{8}$$

where $L$ is the loss function, that is, $L := L(X(T))$. After discretizing the above DDE, we have

$$\begin{aligned} x(t + \Delta t) &= x(t) + \Delta t \cdot f(x(t), y(t), \theta(t)), \\ &= x(t) + \Delta t \cdot f(x(t), x(t - \tau), \theta(t)), \\ x(t + \tau + \Delta t) &= x(t + \tau) + \Delta t \cdot f(x(t + \tau), y(t + \tau), \theta(t + \tau)), \\ &= x(t + \tau) + \Delta t \cdot f(x(t + \tau), x(t), \theta(t + \tau)). \end{aligned} \tag{9}$$

According to the definition of $\lambda(t)$ and applying the chain rule, we have

$$\begin{aligned} \lambda(t) &= \frac{\partial L}{\partial x(t + \Delta t)} \frac{\partial x(t + \Delta t)}{\partial x(t)} + \frac{\partial L}{\partial x(t + \tau + \Delta t)} \frac{\partial x(t + \tau + \Delta t)}{\partial x(t)} \cdot \chi_{[0, T-\tau]}(t) \\ &= \lambda(t + \Delta t) \frac{\partial x(t + \Delta t)}{\partial x(t)} + \lambda(t + \tau + \Delta t) \frac{\partial x(t + \tau + \Delta t)}{\partial x(t)} \cdot \chi_{[0, T-\tau]}(t) \\ &= \lambda(t + \Delta t)(I + \Delta t \cdot f_x(x(t), y(t), \theta(t))) \\ &\quad + \lambda(t + \tau + \Delta t) \Delta t \cdot f_y(x(t + \tau), y(t + \tau), \theta(t + \tau))) \cdot \chi_{[0, T-\tau]}(t) \\ &= \lambda(t + \Delta t) + \Delta t \cdot \left[ \lambda(t + \Delta t) \cdot f_x(t) + \lambda(t + \tau + \Delta t) \cdot f_y(t + \tau) \cdot \chi_{[0, T-\tau]}(t) \right] \end{aligned} \tag{10}$$

which implies,

$$\begin{aligned} \dot{\lambda}(t) &= \lim_{\Delta t \to 0} \frac{\lambda(t + \Delta t) - \lambda(t)}{\Delta t} \\ &= -\lambda(t) \cdot f_x(t) - \lambda(t + \tau) \cdot f_y(t + \tau) \cdot \chi_{[0, T-\tau]}(t), \end{aligned} \tag{11}$$

where $\chi_{[0, T-\tau]}(\cdot)$ is a characteristic function. For the parameter $\theta(t)$, the result can be analogously derived as

$$\begin{aligned} \frac{\partial L}{\partial \theta(t)} &= \frac{\partial L}{\partial x(t + \Delta t)} \frac{\partial x(t + \Delta t)}{\partial \theta(t)} \\ &= \Delta t \cdot \lambda(t + \Delta t) \cdot f_\theta(t) \end{aligned} \tag{12}$$

In this article, $\theta(t)$ is considered to be a constant variable, i.e., $\theta(t) \equiv \theta$, which yields:

$$
\begin{aligned}
\frac{\partial L}{\partial \theta} &= \lim_{\Delta t \to 0} \sum \Delta t \cdot \lambda(t + \Delta t) \cdot f_\theta(t) \\
&= \int_0^T \lambda(t) \cdot f_\theta(t) dt \\
&= \int_T^0 -\lambda(t) \cdot f_\theta(t) dt.
\end{aligned}
\tag{13}
$$

To summarize, we get the gradients with respect to $x(0)$ and $\theta$ in a form of augmented DDEs. These DDEs are backward in time and written in an integral form of

$$
\begin{aligned}
x(0) &= x(T) + \int_T^0 f(x, y, \theta) dt, \\
\frac{\partial L}{\partial x(0)} &= \frac{\partial L}{\partial x(T)} + \int_T^0 -\lambda(t) \cdot f_x(t) - \lambda(t + \tau) \cdot f_y(t + \tau) \cdot \chi_{[0, T-\tau]}(t) dt, \\
\frac{\partial L}{\partial \theta} &= \int_T^0 -\lambda(t) \cdot f_\theta(t) dt.
\end{aligned}
\tag{14}
$$

### A.2   THE SECOND PROOF OF THEOREM 1

Here, we mainly employ the Lagrangian multiplier method and the calculus of variation to derive the adjoint method for the NDDEs. First, we define the Lagrangian by

$$
\mathcal{L} := L(X(T)) + \int_0^T \lambda(t)(\dot{x} - f(x(t), y(t), \theta)) dt.
\tag{15}
$$

We thereby need to find the so-called Karush-Kuhn-Tucker (KKT) conditions for the Lagrangian, which is necessary for finding the optimal solution of $\theta$. To obtain the KKT conditions, the calculus of variation is applied by taking variations with respect to $\lambda(t)$ and $x(t)$.

For $\lambda(t)$, let $\hat{\lambda}(t)$ be a continuous and differentiable function with a scalar $\epsilon$. We add the perturbation $\hat{\lambda}(t)$ to the Lagrangian $\mathcal{L}$, which results in a new Lagrangian, denoted by

$$
\mathcal{L}(\epsilon) := L(x(T)) + \int_0^T (\lambda(t) + \epsilon \hat{\lambda}(t))(\dot{x} - f(x(t), y(t), \theta)) dt.
\tag{16}
$$

In order to obey the optimal condition for $\lambda(t)$, we require the following equation

$$
\frac{d\mathcal{L}(\epsilon)}{d\epsilon} = \int_0^T \hat{\lambda}(t)(\dot{x} - f(x(t), y(t), \theta)) dt
\tag{17}
$$

to be zero. Due to the arbitrariness of $\hat{\lambda}(t)$, we obtain

$$
\dot{x} - f(x(t), y(t), \theta) = 0, \quad \forall t \in [0, T],
\tag{18}
$$

which is exactly the DDE forward in time.

Analogously, we can take variation with respect to $x(t)$ and let $\hat{x}(t)$ be a continuous and differentiable function with a scalar $\epsilon$. Here, $\hat{x}(t) = 0$ for $t \in [-\tau, 0]$. We also denote by $\hat{y}(t) := \hat{x}(t - \tau)$ for $t \in [0, T]$. The new Lagrangian under the perturbation of $\hat{x}(t)$ becomes

$$
\mathcal{L}(\epsilon) := L(x(T) + \epsilon \hat{x}(T)) + \int_0^T \lambda(t) \left( \frac{dx(t) + \epsilon \hat{x}(t)}{dt} - f(x(t) + \epsilon \hat{x}(t), y(t) + \epsilon \hat{y}(t), \theta) \right) dt.
\tag{19}
$$

We then compute the $\frac{d\mathcal{L}(\epsilon)}{d\epsilon}$, which gives

$$\frac{d\mathcal{L}(\epsilon)}{d\epsilon}\Big|_{\epsilon=0} = \frac{\partial L}{\partial x(T)}\hat{x}(T) + \int_0^T \lambda(t)\left(\frac{d\hat{x}(t)}{dt} - f_x(x(t), y(t), \theta)\hat{x}(t) - f_y(x(t), y(t), \theta)\hat{y}(t)\right)dt$$

$$= \frac{\partial L}{\partial x(T)}\hat{x}(T) + \lambda(t)\hat{x}(t)|_0^T + \int_0^T -\hat{x}(t)\frac{d\lambda(t)}{dt} \qquad \text{integration by parts}$$

$$+ \int_0^T -\lambda(t)f_x(x(t), y(t), \theta)\hat{x}(t) - \lambda(t)f_y(x(t), y(t), \theta)\hat{y}(t)dt$$

$$= \left(\frac{\partial L}{\partial x(T)} + \lambda(T)\right)\hat{x}(T) + \int_0^T -\hat{x}(t)\frac{d\lambda(t)}{dt} - \lambda(t)f_x(x(t), y(t), \theta)\hat{x}(t)$$

$$- \int_0^T \lambda(t)f_y(x(t), y(t), \theta)\hat{x}(t - \tau)dt$$

$$= \left(\frac{\partial L}{\partial x(T)} + \lambda(T)\right)\hat{x}(T) + \int_0^T -\hat{x}(t)\frac{d\lambda(t)}{dt} - \lambda(t)f_x(x(t), y(t), \theta)\hat{x}(t)$$

$$- \int_{-\tau}^{T-\tau} \lambda(t' + \tau)f_y(x(t' + \tau), y(t' + \tau), \theta)\hat{x}(t')dt'$$

$$= \left(\frac{\partial L}{\partial x(T)} + \lambda(T)\right)\hat{x}(T) + \int_0^T -\hat{x}(t)\frac{d\lambda(t)}{dt} - \lambda(t)f_x(x(t), y(t), \theta)\hat{x}(t)$$

$$- \int_0^{T-\tau} \lambda(t')f_y(x(t' + \tau), y(t' + \tau), \theta)\hat{x}(t')dt' \qquad \text{no variation on the interval } [-\tau, 0]$$

$$= \left(\frac{\partial L}{\partial x(T)} + \lambda(T)\right)\hat{x}(T)$$

$$+ \int_0^T \hat{x}(t)\left(-\frac{d\lambda(t)}{dt} - \lambda(t)f_x(x(t), y(t), \theta) - \lambda(t + \tau)f_y(x(t + \tau), y(t + \tau), \theta)\chi_{[0,T-\tau]}(t)\right)dt \tag{20}$$

Notice that $\frac{d\mathcal{L}(\epsilon)}{d\epsilon}\big|_{\epsilon=0} = 0$ is satisfied for all continuous differentiable $\hat{x}(t)$ at the optimal $x(t)$. Thus, we have

$$\frac{d\lambda(t)}{dt} = -\lambda(t)f_x(x(t), y(t), \theta) - \lambda(t + \tau)f_y(x(t + \tau), y(t + \tau), \theta)\chi_{[0,T-\tau]}(t), \quad \lambda(T) = \frac{\partial L}{\partial x(T)}. \tag{21}$$

Therefore, the adjoint state follows a DDE as well.

### A.3  THE PROOF OF THEOREM 2

The validation for Theorem 2 is straightforward. Consider the NDDEs in the following form:

$$\begin{cases} \frac{d\boldsymbol{h}_t}{dt} = f(\boldsymbol{h}_{t-\tau}; \boldsymbol{w}), & t >= 0, \\ \boldsymbol{h}(t) = \boldsymbol{x}, & t <= 0 \end{cases}$$

where $\tau$ equals to the final time $T$. Due to $\boldsymbol{h}(t) = \boldsymbol{x}$ for $t \leq 0$, then the vector field of the NDDEs in the interval $[0, T]$ is constant. This implies $\boldsymbol{h}(T) = \boldsymbol{x} + T \cdot f(\boldsymbol{x}; \boldsymbol{w})$. Assume that the neural network $f(\boldsymbol{x}; \boldsymbol{w})$ is able to approximate the map $G(\boldsymbol{x}) = \frac{1}{T}[F(\boldsymbol{x}) - \boldsymbol{x}]$. Then, we have $\boldsymbol{h}(T) = \boldsymbol{x} + T \cdot \frac{1}{T}[F(\boldsymbol{x}) - \boldsymbol{x}] = F(\boldsymbol{x})$. The proof is completed.

### A.4  COMPLEXITY ANALYSIS OF THE ALGORITHM 1

As shown in (Chen et al., 2018), the memory and the time costs for solving the NODEs are $\mathcal{O}(1)$ and $\mathcal{O}(L)$, where $L$ is the number of the function evaluations in the time interval $[0, T]$. More precisely, solving the NODEs is memory efficient without storing any intermediate states of the evaluated time points in solving the NODEs. Here, we intend to analyze the complexity of Algorithm 1. It should be noting that the state-of-the-art for the DDE software is not as advanced as that for the ODE software. Hence, solving a DDE is much difficult compared with solving the ODE. There exists several DDE

solvers, such as the popular solver, the dde23 provided by MATLAB. However, these DDE solvers usually need to store the history states to help the solvers access the past time state $h(t - \tau)$. Hence, the memory cost of DDE solvers is $\mathcal{O}(H)$, where $H$ is the number of the history states. This is the major difference between the DDEs and the ODEs, as solving the ODEs is memory efficient, i.e., $\mathcal{O}(1)$. In Algorithm 1, we propose a piece-wise ODE solver to solve the DDEs. The underlying idea is to transform the DDEs into the piece-wise ODEs for every $\tau$ time periods such that one can naturally employ the framework of the NODEs. More precisely, we compute the state at time $k\tau$ by solving an augmented ODE with the augmented initial state, i.e., concatenating the states at time $-\tau, 0, \tau, ..., (k-1)\tau$ into a single vector. Such a method has several strengths and has weaknesses as well. The strengths include:

- One can easily implement the algorithm using the framework of the NODEs, and

- Algorithm 1 becomes quite memory efficient $\mathcal{O}(n)$, where we only need to store a small number of the forward states, $\boldsymbol{h}(0), ..., \boldsymbol{h}(n\tau)$, to help the algorithm compute the adjoint and the gradients in a reverse mode. Here, the final $T$ is assumed to be not very large compared with the time delay $\tau$, for example, $T = n\tau$ with a small integer $n$.

Notably, we chosen $n = 1$ (i.e., $T = \tau = 1.0$) of the NDDEs in the experiments on the image datasets, resulting in a quite memory efficient computation. Additionally, the weaknesses involve:

- Algorithm 1 may suffer from a high memory cost, if the final $T$ is extremely larger than the $\tau$ (i.e., many forward states, $\boldsymbol{h}(0), ..., \boldsymbol{h}(n\tau)$, are required to be stored), and

- the increasing augmented dimension of the ODEs may lead to the heavy time cost for each evaluation in the ODE solver.

## B EXPERIMENTAL DETAILS

### B.1 SYNTHETIC DATASETS

For the classification of the dataset of concentric spheres, we almost follow the experimental configurations used in (Dupont et al., 2019). The dataset contain 2000 data points in the outer annulus and 1000 data points in the inner sphere. We choose $r_1 = 0.5$, $r_2 = 1.0$, and $r_3 = 1.5$. We solve the classification problem using the NODE and the NDDE, whose structures are described, respectively, as:

1. Structure of the NODEs: the vector field is modeled as

$$f(\boldsymbol{x}) = W_{out}\text{ReLU}(W\text{ReLU}(W_{in}\boldsymbol{x})),$$

where $W_{in} \in \mathbb{R}^{32 \times 2}$, $W \in \mathbb{R}^{32 \times 32}$, and $W_{out} \in \mathbb{R}^{2 \times 32}$;

2. Structure of the NDDEs: the vector field is modeled as

$$f(\boldsymbol{x}(t - \tau)) = W_{out}\text{ReLU}(W\text{ReLU}(W_{in}\boldsymbol{x}(t - \tau))),$$

where $W_{in} \in \mathbb{R}^{32 \times 2}$, $W \in \mathbb{R}^{32 \times 32}$, and $W_{out} \in \mathbb{R}^{2 \times 32}$.

For each model, we choose the optimizer Adam with 1e-3 as the learning rate, 64 as the batch size, and 5 as the number of the training epochs.

We also utilize the synthetic datasets to address the regression problem of the time series. The optimizer for these datasets is chosen as the Adam with $0.01$ as the learning rate and the mean absolute error (MAE) as the loss function. In this article, we choose the true time delays of the underlying systems for the NDDEs. We describe the structures of the models for the synthetic datasets as follows.

For the DDE $\dot{\boldsymbol{x}} = \boldsymbol{A}\tanh(\boldsymbol{x}(t) + \boldsymbol{x}(t - \tau))$, we generate a trajectory by choosing its parameters as $\tau = 0.5$, $A = [[-1, 1], [-1, -1]]$, $\boldsymbol{x}(t) = [0, 1]$ for $t \leq 0$, the final time $T = 2.5$, and the sampling time period equal to $0.1$. We model it using the NODE and the NDDE, whose structures are illustrated, respectively, as:

1. Structure of the NODE: the vector field is modeled as

$$f(x) = W_{out} \tanh(W_{in}x),$$

where $W_{in} \in \mathbb{R}^{10 \times 2}$ and $W_{out} \in \mathbb{R}^{2 \times 10}$;

2. Structure of the NDDE: the vector field is modeled as

$$f(x(t), x(t-\tau)) = W_{out} \tanh(W_{in}(x(t) + x(t-\tau))),$$

where $W_{in} \in \mathbb{R}^{10 \times 2}$ and $W_{out} \in \mathbb{R}^{2 \times 10}$.

We use the whole trajectory to train the models with the total iterations equal to 5000.

For the population dynamics $\dot{x} = rx(t)(1 - x(t-\tau_1))$ with $x(t) = x_0$ for $t \leq 0$ and the Mackey-Glass systems $\dot{x} = \beta \frac{x(t-\tau)}{1+x^n(t-\tau_2)} - \gamma x(t)$ with $x(t) = x_0$ for $t \leq 0$, we choose the parameters as $\tau_1 = 1$, $\tau_2 = 1$, $r = 1.8$, $\beta = 4$, $n = 9.65$, and $\gamma = 2$. We then generate 100 different trajectories within the time interval $[0, 8]$ with 0.05 as the sampling time period for both the population dynamics and the Mackey-Glass systems. We split the trajectories into two parts. The first part within the time interval $[0, 3]$ is used as the training data. The other part is used for testing. We model both systems by applying the NODE, the NDDE, and the ANODE, whose structures are introduced, respectively, as:

1. Structure of the NODE: the vector field is modeled as

$$f(x) = W_{out} \tanh(W \tanh(W_{in}x)),$$

where $W_{in} \in \mathbb{R}^{10 \times 1}$, $W \in \mathbb{R}^{10 \times 10}$, and $W_{out} \in \mathbb{R}^{1 \times 10}$;

2. Structure of the NDDE: the vector field is modeled as

$$f(x(t), x(t-\tau)) = W_{out} \tanh(W \tanh(W_{in}\text{concat}(x(t), x(t-\tau)))),$$

where $W_{in} \in \mathbb{R}^{10 \times 2}$, $W \in \mathbb{R}^{10 \times 10}$, and $W_{out} \in \mathbb{R}^{1 \times 10}$.

3. Structure of the ANODE: the vector field is modeled as

$$f(\boldsymbol{x}_{aug}(t)) = W_{out} \tanh(W \tanh(W_{in}\boldsymbol{x}_{aug})),$$

where $W_{in} \in \mathbb{R}^{10 \times 2}$, $W \in \mathbb{R}^{10 \times 10}$, $W_{out} \in \mathbb{R}^{2 \times 10}$, and the augmented dimension is equal to 1. Notably, we need to align the augmented trajectories with the target trajectories to be regressed. To do it, we choose the data in the first component and exclude the data in the augmented component, i.e., simply projecting the augmented data into the space of the target data.

We train each model for the total 3000 iterations.

## B.2 IMAGE DATASETS

The image experiments shown in this article are mainly depend on the code provided in (Dupont et al., 2019). Also, the structures of each model almost follow from the structures proposed in (Dupont et al., 2019). More precisely, we apply the convolutional block with the following strutures and dimensions

- $1 \times 1$ conv, $k$ filters, 0 paddings,
- ReLU activation function,
- $3 \times 3$ conv, $k$ filters, 1 paddings,
- ReLU activation function,
- $1 \times 1$ conv, $c$ filters, 0 paddings,

where $k$ is different for each model and $c$ is the number of the channels of the input images. In Tab. 2, we specify the information for each model. As can be seen, we fix approximately the same number of the parameters for each model. We select the hyperparamters for the optimizer Adam through 1e-3 as the learning rate, 256 as the batch size, and 30 as the number of the training epochs.

## C ADDITIONAL RESULTS FOR THE EXPERIMENTS

|  | CIFAR10 | MNIST | SVHN |
|---|---|---|---|
| NODE | 92/107645 | 92/84395 | 92/107645 |
| NDDE | 92/107921 | 92/84487 | 92/107921 |
| NODE+NDDE | 64/105680 | 64/82156 | 64/105680 |
| A1+NODE | 86/108398 | 87/84335 | 86/108398 |
| A1+NDDE | 85/107189 | 86/82944 | 85/107189 |
| A1+NODE+NDDE | 60/107218 | 61/83526 | 60/107218 |
| A2+NODE | 79/108332 | 81/83230 | 79/108332 |
| A2+NDDE | 78/107297 | 81/83473 | 78/107297 |
| A2+NODE+NDDE | 55/107265 | 57/83101 | 55/107265 |
| A4+NODE | 63/108426 | 70/84155 | 63/108426 |
| A4+NDDE | 62/107719 | 69/83237 | 62/107719 |
| A4+NODE+NDDE | 43/106663 | 49/83859 | 43/106663 |

Table 2: The number of the filters and the whole parameters in each model used for CIFAR10, MNIST, and SVHN. In the first column, A$p$ with $p = 1, 2, 4$ indicates the augmentation of the image space $\mathbb{R}^{c \times h \times w} \rightarrow \mathbb{R}^{(c+p) \times h \times w}$.

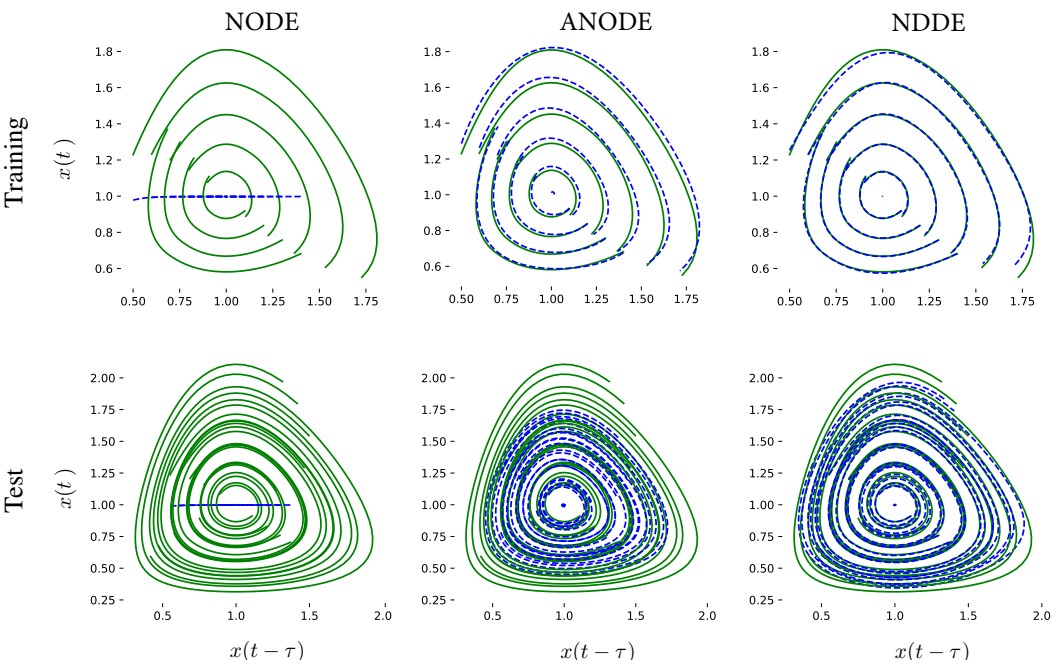

Figure 9: The phase portraits of the population dynamics in the training and the test stages. We only show the 10 phase portraits from the total 100 phase portraits for the training and the testing time-series. Here, the solid lines correspond to the true dynamics, while the dash lines correspond to the trajectories generated by the trained models in the training duration and in the test duration, respectively.

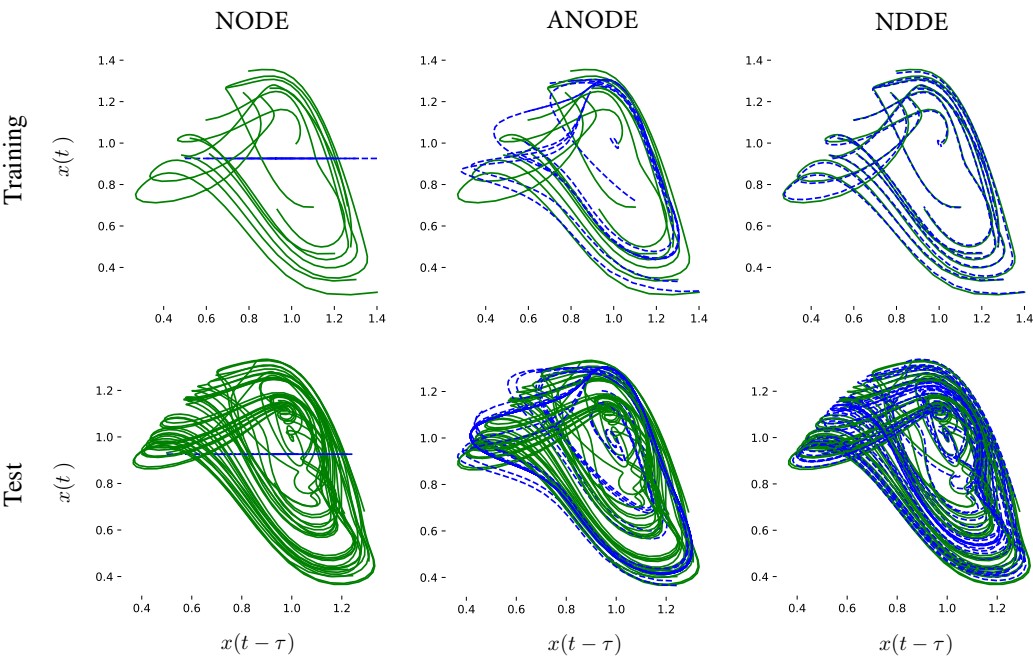

Figure 10: The phase portraits of the Makey-Glass systems exhibiting chaotic dynamics in the training and the test stages. We only present the 10 phase portraits from the total 100 phase portraits for the training and the testing time-series. Here, the solid lines correspond to the true dynamics, while the dash lines correspond to the trajectories generated by the trained models in the training duration and in the test duration, respectively.

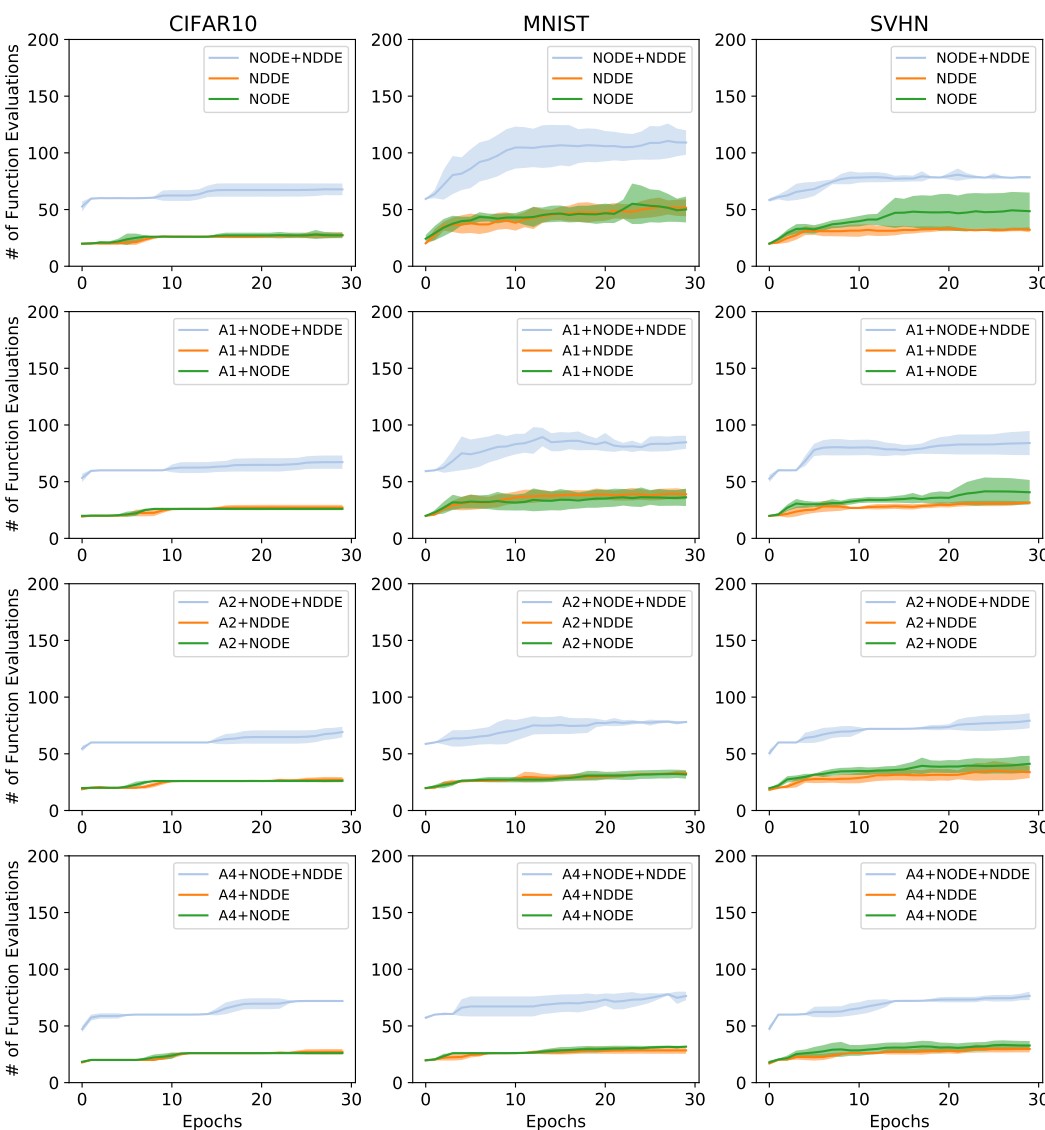

Figure 11: Evolution of the forward number of the function evaluations (NFEs) during the training process for each model on the image datesets, CIFAR10, MNIST, and SVHN.

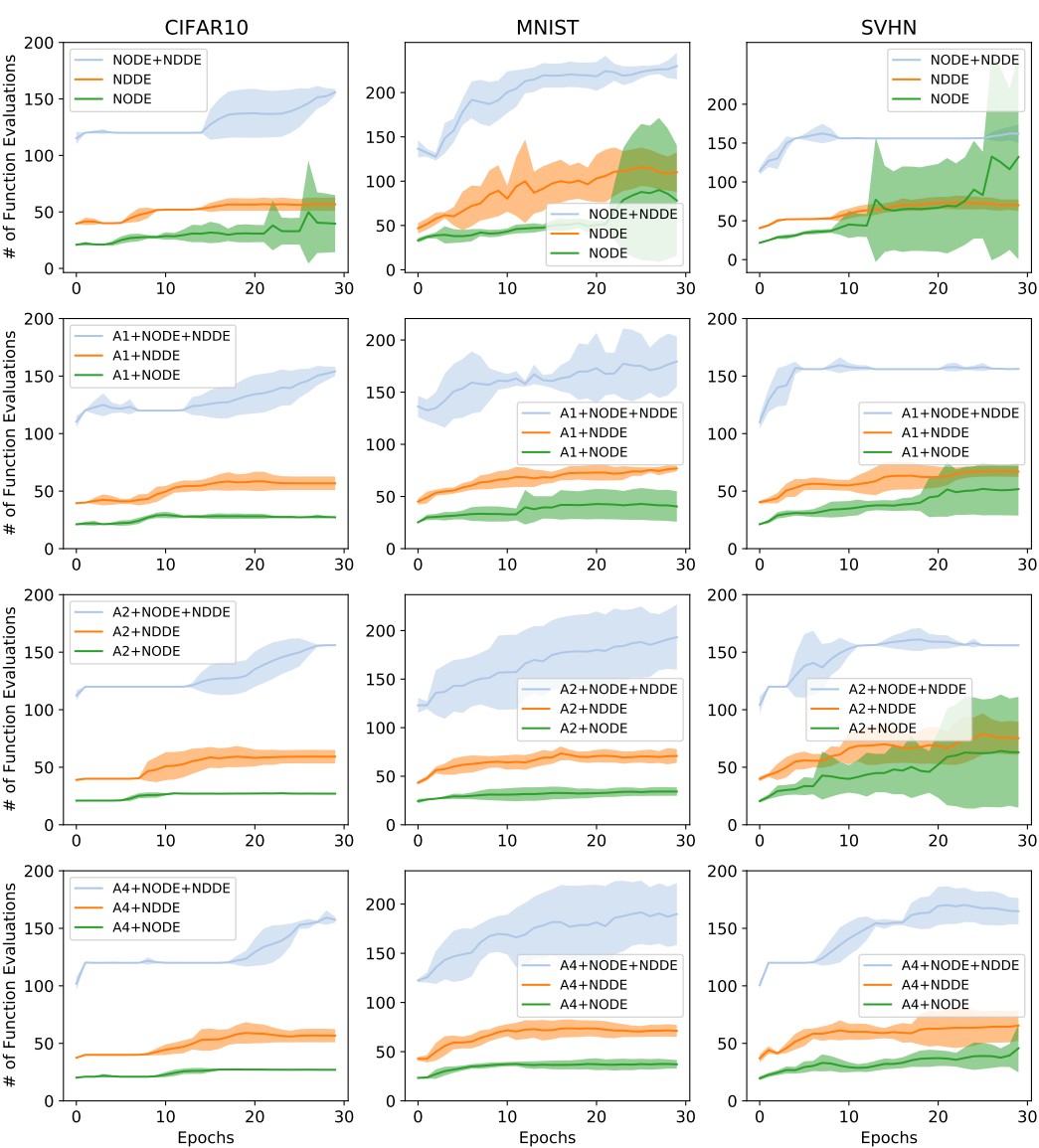

Figure 12: Evolution of the backward number of the function evaluations (NFEs) during the training process for each model on the image datesets, CIFAR10, MNIST, and SVHN.

