# OpenReview forum: "Neural Delay Differential Equations"
_ICLR.cc/2021/Conference — ICLR 2021 Poster_

### Official Review · AnonReviewer2 · 2020-10-28
**Interesting model class, improper characterization and comparison to prior work**

**Rating:** 6
**Confidence:** 4

**Review:**

This paper presents a modeling class of parameterized first-order
differential equations that are conditional on some delayed
past states. This is a promising direction for the community to go
to push past current ODE modeling limitations, but I recommend for
rejection in the current form due to improper characterization
and evaluation with respect to prior work (more details below).

# Strengths
To the best of my knowledge this is a novel modeling class
for neural ODEs. Delay ODEs are well-studied in contexts
outside of the machine learning community that would be
useful to bring into it.

On the representation capacity, delay ODEs are able to
overcome one instance of an intersecting path issue that
the usual first-order neural ODEs are unable to represent.

The application to population dynamics and Mackey-Glass systems
that are naturally represented an delay differential equations
seems reasonable and well-motivated that baseline methods may
not be able to capture this modeling class.

# Weaknesses
The strongest weaknesses of this paper are the characterization
and evaluation in comparison to [Dupont 2019], which points
out some of the same modeling issues and proposes an
alternative way to fix them.
Even though this paper cites [Dupont2019], it omits that they consider
identical experimental settings and does not include their results:

  1. Figure 4 and 5 are almost identical to figures in [Dupont2019]

  2. Table 1 presents results in MNIST and CIFAR10 and is identical
     to the setting in [Dupont2019]. The neural ODE baselines
     are almost consistent with the ones reported in [Dupont2019], but
     [Dupont2019] outperforms the method presented here, and are
     omitted from the table. [Dupont2019] additionally
     considers the SVHN dataset.

# Other comments and questions
Is the adjoint method for the NDDEs very different than the adjoint
method for NODEs? It could be insightful to intuitively explain
this.

The extension to more general delay differential equations seems
promising and working out the details in this general case in
addition to the case of having a single delay would add to the paper's
theory and methodological contributions.

---

> ### Author Response · Authors · 2020-11-21
> **Thank you for your comments**
>
> We would like to thank the reviewer for the comments and valuable suggestions. For the main changes in the revised paper, please refer to the “**Response to all reviewers**”.
> ```
> Q1: The strongest weaknesses of this paper are the characterization and evaluation in comparison to [Dupont 2019], which points out some of the same modeling issues and proposes an alternative way to fix them. Even though this paper cites [Dupont2019], it omits that they consider identical experimental settings and does not include their results.
> ```
> **Response**: Many thanks for your careful reading and valuable comments. We agree with you that the previous paper was not in a proper way for the characterization and evaluation in comparison to [Dupont 2019]. We thereby try our best to make a fair comparison to the great work [Dupont 2019]. Please refer to the Figure 8 and Table 1 in the revised paper, which reveal the good modeling capabilities and feature representations of the NDDEs. The comparisons are done under the same augmented dimension and using the A+NODE (equivalently, ANODE), A+NDDE, and A+NODE+NDDE.
> ```
> Q2: Figure 4 and 5 are almost identical to figures in [Dupont2019]
> ```
> **Response**:  Many thanks for your comments. We do generate the plots of the NODEs in Figures 4 and 5.  We thereby added the additional captions for the Figures 4 and 5, also citing the paper [Dupont2019].
> ```
> Q3: Is the adjoint method for the NDDEs very different than the adjoint method for NODEs? It could be insightful to intuitively explain this.
> ```
> **Response**: Many thanks for this question. As can be seen in the Theorem 1, the dynamics of adjoint of the NDDEs is another DDE while the adjoint dynamics of the NODEs follows an ODE. We think such a fact is the major difference between the NODEs and the NDDEs. This definitely leads to a more complex but intrincially different algorithm for solving the adjoint dynamics as shown in Algorithm 1.
> ```
> Q4: The extension to more general delay differential equations seems promising and working out the details in this general case in addition to the case of having a single delay would add to the paper's theory and methodological contributions.
> ```
> **Response**:  Many thanks for your valuable comments. As we discussed in the previous version, we do believe that the extensions of the NDDEs could be more promising. We will further consider these general cases in the future research directions. In the revised paper, we do make the efforts to extend the NDDEs, such as the NODE+NDDE, which models the initial function of the NDDEs as an ODE.
>
> We would like to thank the reviewer again for your valuable comments and thoughtful suggestions. We thereby improve the paper not only from the theoretical aspect but also the experimental aspect. We hope that the revised paper have been much improved. We do hope that the response and the revised paper, especially the added experiments with the corresponding descriptions, have addressed the main concerns fully.  We would appreciate it very much if you could support our work and we are willing to make further improvements with any constructive feedback from your side.

---

> > ### Comment · AnonReviewer2 · 2020-11-24
> > **Updated score to WA, the revised paper addresses my concerns**
> >
> > Many thanks for the clarifications and revisions. The revised paper reads nicely and this addresses my concerns on the positioning with respect to [Dupont2019].

---

> > > ### Author Response · Authors · 2020-11-25
> > > **Re: Updated score to WA, the revised paper addresses my concerns**
> > >
> > > Thank you very much!

---

> > > > ### Author Response · Authors · 2020-11-25
> > > > **Re: Updated score to WA, the revised paper addresses my concerns**
> > > >
> > > > Thank you very much for your comments.  We'd appreciate your positive support to our works.

---

### Official Review · AnonReviewer1 · 2020-10-28
**An interesting  and promising extension of NODEs that leverages Delay Differential Equations but would benefit from further study**

**Rating:** 5
**Confidence:** 4

**Review:**

This paper contributes to a recent line of researches linked to Neural Ordinary Differential Equations (Chen et al. 2018) and variants.
This new family of deep neural networks (NODE) generalize ideas from Residual Networks and consider continuous dynamics of hidden units using an Ordinary Differential Equation (ODE) specified by a neural network. Computing in such networks consists in taking x=h(0) as input and define the output layer h(T) to be the solution of an ODE initial value problem at time T. The main ingredient which makes learning possible is the backpropagation algorithm through the last layer ODE solver, that relies on the adjoint sensitivity method (Pontryagin et al., 1962).

This work considers Delay Differential Equations instead of ODE, which allows to implement more complex dynamics and thus achieve estimation of more complex functions. The contributions of the paper are the following:
(1)	a novel deep continuous -time deep neural network defined by the following equations
For a given delay \tau (never discussed), Neural Delay Differential equations define dynamics of the form:
dh_t\dt = f(h_t,h_{t-\tau}, t ;w),  t \geq 0,
h(t) = \phi(t), t \leq 0
        In this way the network can take into account a former hidden layer.
(2)	the derivation of the adjoint sensitivity method for delayed equations, which seems relatively straightforward and which is backed by two proofs in the appendix.
(3)	A novel learning algorithm that implements the forward for h and the backward pass for h,lambda (the augmented variable) and dL/dw (L is the loss) by a piece-wise ODE solver, dealing with the different delayed states.
(4)	Experiments on 2D toydatasets and on classic differential equations such as Mackey-Glass are shown to exhibit the ability of DDE to cope with those dynamics, in constrast to ODE.
(5)	Experiments on image datasets
(6)
The strong point of this paper is of course the proposal of the new variant of NODE, which comes with a novel algorithm and overcomes some limitations of NODE. I was interested by the examples of functions not covered by NODE and covered by NDDEand easily convinced by that.

There are some weak points in this work. Some of them could be easily improved I think, others call for further  work.
Relatively Minor points
-	The paper is not self-content and does not a very good job in explaining the context of Neural ODE. I suggest that you more clearly describe NODE, as a hypothesis class and then as a learning algorithm.
-	The two uses of NDDE concern modeling time-series  or implementing a classification/regression function. Can you each time precise inputs/outputs, samples you use in training. Just for clarity, the reader guesses of course but its rather slows down the reading this absence of notation and formalization ?
Major points and questions to the authors:
-	More importantly, as a novel algorithm is introduced, I expect to see a complexity in time analysis. As for NODE I understand that the complexity in space is favorable.
-	An associated question is the role of tau, the delay. How is it chosen ? I imagine that if tau converges towards 0 we find the behaviour of NODE ? what was its value in the two real-world experiments? Was it selected by cross-validation ?  Is it too computationally heavy to consider multiple delays ?
-	Eventually, I have doubts and questions about the real opportunities for using NDDE in the two real word datasets: yes the divergence of NODE poses problems : in these experiments no doubt that NDDE has a more stable empirical behavior, with a small variance. However, in fine, the average performance is nearly of the same order, a significant difference only on MNIST but with a very bad score considering this is an easy problem. I won’t qualify these results as exceptional and I kindly engage the authors to remain modest.
-	Now can we cope without delays by introducing new states in the way augmented  Neural ODE are built (Dupont et al.) ?
-	For what kinds of classification problems NDDE could be more interesting than NODE – The question is certainly difficult.

In conclusion, the paper tackles a very interesting topic and I had pleasure to discover it (with the help of literature). Although the idea to consider DDE instead of ODE is incremental, its relevant and novel, and certainly promising and worth to be explored because it addresses some of the issues of NODE.
However I stay on my hunger on different points, there are pending questions that require to be answered before acceptance.

---

> ### Author Response · Authors · 2020-11-21
> **Thank you for your comments (1/2)**
>
> We thanks for your overall positive feedback, valuable comments and helpful suggestions on the NDDEs. We first recommend the reviewer to read the “**Response to all reviewers**”. For the minor/major points on this work, we response them one by one. Hopefully, the following answers could help you understand our work better.
> ```
> Q1: The paper is not self-content and does not a very good job in explaining the context of Neural ODE. I suggest that you more clearly describe NODE, as a hypothesis class and then as a learning algorithm.
> ```
> **Response**: Thanks for this comments. We made the efforts to explain the context of the Neural ODEs in the revised paper.
> ```
> Q2: The two uses of NDDE concern modeling time-series or implementing a classification/regression function. Can you each time precise inputs/outputs, samples you use in training. Just for clarity, the reader guesses of course but its rather slows down the reading this absence of notation and formalization?
> ```
> **Response**: Many thanks for your comments. For the regression problems,  its adjoint dynamics is different from the one of the classification problem. Because, for the regression task of the time series, the loss function may directly depend on the state at multiple observation times, such as of the form $L(h(t_0), h(t_1),...,h(t_n))=\Sigma_i (h(t_i) - h_{target}(t_i))^2$, while for the classification problem, we assume the loss function in the form of $L(h(T))$. Under such a case, we should update the adjoint state instantly by adding the partial derivative of the loss at each observational time point. To make it more clearly, we made the additional efforts to illustrate the difference between the classification and the regression problems in the revised paper (please see the end of  "NODEs" of the Second part "Related works").
> ```
> Q3: More importantly, as a novel algorithm is introduced, I expect to see a complexity in time analysis. As for NODE I understand that the complexity in space is favorable.
> ```
> **Response**: Many thanks for your comments. As pointed out in the previous version, the algorithm may suffer a high computational cost. This is because we solve the NDDEs including the reverse-mode of the gradients by a piece-wise ODE solver. We provided the analysis of the complexity of the algorithm in the Appendix.
> ```
> Q4: An associated question is the role of tau, the delay. How is it chosen? I imagine that if tau converges towards 0 we find the behavior of NODE? what was its value in the two real-world experiments? Was it selected by cross-validation? Is it too computationally heavy to consider multiple delays?
> ```
> **Response**: Many thanks for your comments. We agree with you that if tau converges towards $0$ we find the behavior of NODE. In the real-world experiments. we simply choose the delay $\tau=1.0$ and the final time $T=1.0$ as well. However, under such a simple setting, the NDDEs and its extensions can achieve a quite good performance on these image datasets, including the SVHN dataset in the revised paper. As discussed in the paper, considering the multiple delays could be the future direction of the NDDEs. Notably, as you pointed out, the multiple delays may result in a heavy computational cost.
> ```
> Q5: Eventually, I have doubts and questions about the real opportunities for using NDDE in the two real word datasets: yes the divergence of NODE poses problems: in these experiments no doubt that NDDE has a more stable empirical behavior, with a small variance. However, in fine, the average performance is nearly of the same order, a significant difference only on MNIST but with a very bad score considering this is an easy problem. I won’t qualify these results as exceptional and I kindly engage the authors to remain modest.
> ```
> **Response**: Many thanks for this advice. We agree with you that we should remain modest on the description of the results of the experiments. Moreover, to convince the reviewers, we further conducted additional experiments, including a new dataset (SVHN), and the extensions of the NDDEs (NODE+NDDE, A+NDDE, A+NODE+NDDE). As is expected, the NDDEs and its variants eventually obtained a quite good performance on the real-word datasets. For more details, please refer to the “**Response to all reviewers**” and the revised paper.

---

> > ### Author Response · Authors · 2020-11-21
> > **Thank you for your comments (2/2)**
> >
> > ```
> > Q6: Now can we cope without delays by introducing new states in the way augmented Neural ODE are built (Dupont et al.)?
> > ```
> > **Response**:  Many thanks for your insightful question. Now, we cannot confirm that we can cope without delays by introducing new states in the way augmented Neural ODE are built (Dupont et al.). Here, we provide out insights on this question. Notably, for the NDDEs, we need to define an infinitely-dimensional set of initial conditions in the interval $[-\tau, 0]$ [1].  That is, the NDDEs could be regarded as the infinitely-dimensional system while the NODEs and ANODEs are finite-dimension system. From such a viwpoint, we may not model the intrinsic dynamic of the underlying time-delayed systems by the ANODEs  (see the Figure 7).  Anyway, this question is interesting, which needs more deeper analytical investigations.
> > ```
> > Q7: For what kinds of classification problems NDDE could be more interesting than NODE – The question is certainly difficult.
> > ```
> > **Response**: Many thanks for your comments. This is absolutely a good question. From our point view, we believe that the NDDEs may be a proper model to reveal the true dynamics of the underlying system with the delay. Moreover, as the shown the revised paper, the NDDEs and its extensions achieve the good performance on the classifications of the three image datasets. We believe that the NDDEs can be applied to other large image datasets but may suffer the heavy computational cost.
> >
> > We thank the reviewer again for your valuable comments. We do believe that the quality of the revised paper has been improved eceptionally from the experimental and theoretical aspects thanks to the reviewers’ comments and suggestions.  Hopefully, the responses and the revised paper have sufficiently addressed the main concerns and then the reviewer will reconsider the assessment in support of the revised paper. We are looking forward to your feedback to make further improvements of the paper.
> >
> > [1]. Thomas Erneux. Applied delay differential equations, volume 3. Springer Science & Business Media, 2009.

---

### Official Review · AnonReviewer4 · 2020-10-29
**interesting idea, nice illustrations, preliminary expts, not much theoretical support**

**Rating:** 6
**Confidence:** 3

**Review:**

-- The motivation for resorting to DDEs over ODEs is laid out reasonably well, although I think the intuition behind why NODEs cannot learn certain functions is lacking (specifically, that trajectories cannot cross due to the lack of "memory" provided by the delay).

-- Figure 2 could be more informative. As it stands, it only indicates that the initial state of a NDDE is specified by a function over an interval, rather than a single vector, which is all that is being used.

-- Algorithm 1 is challenging to understand without some knowledge of how DDEs are solved.

-- In the experiments, NDDEs are only compared against NODEs, but it seems like ANODEs are the real competition. Notably in the image experiments performed in the ANODE paper they report substantially higher accuracies than NDDEs.

-- This is mentioned in the discussion section, but I think the paper would benefit from an experiment involving a real-world dataset with some sort of delay component, ideally in which NDDEs outperform NODEs/ANODEs. The existing synthetic experiments are designed to show the strengths of NDDEs, while the image classification tasks don't substantially differentiate them from NODEs.

-- The number of function evaluations is not reported anywhere, which seems like a significant statistic when NDDEs are substantially more complex than NODEs.

-- Figure 7 could probably be improved by reducing the number of different parameter values reported (e.g. 3 columns instead of 6). The description also lists 5 parameter values but there are 6 columns.

-- The discussion paragraph "Extensions of the NDDEs" suggests a generalization to more complex delays, but all of the models in this paper ensure that the delayed value is always the constant initial value. Also, modeling the initial function as an ODE is mentioned, and I wonder whether the implication is that this would involve learning the initial function as a NODE itself. Either way, it's not obvious how either of these would yield better representations.

---

> ### Author Response · Authors · 2020-11-21
> **Thank you for your comments (1/1)**
>
> We thank the reviewer for the overall positive feedback and the valuable comments. We recommend the reviewer to read the “**Response to all reviewers**”. We hope the reviewer could find the efforts and the improvements we made not only from the theoretical aspect and the experimental aspect. For the individual comments, we are going to response them one by one.
> ```
> Q1: The motivation for resorting to DDEs over ODEs is laid out reasonably well, although I think the intuition behind why NODEs cannot learn certain functions is lacking (specifically, that trajectories cannot cross due to the lack of "memory" provided by the delay).
> ```
> **Response**: Thanks for your comments. The certain functions that cannot be modeled by NODEs have been reported in [1], whose proof is based on the fools from the ODE theory and the topology. They provided the intuition behind as well. The intuition behind is simple.  Two distinguish trajectories generated by an autonomous ODE cannot intersect each other; otherwise, this violates the uniqueness of the solution for an ODE. If we want to enforce the NODEs to learn the function $g_{1-D}(x)$, the trajectories from the initial point 1 and -1 must cross each other. That is why the NODEs cannot represent the $g_{1-D}(x)$ [1].  As for the more general class of the function $g(x)$ shown in the paper, Dupont et. al. [1] also provide a similar intuition for the limitations of the modeling capabilities of NODEs.   Since we hope that the flow of NODEs can eventually linearly separate two regions of $g(x)$ while the one region is enclosed by the other region, which thus requires the trajectories to intersect; however, this is impossible.  Moreover, it is worth noting that we further provide a theorem in the revised paper, showing that the NDDEs are of universal approximators.  Acutally, the trajectories generated by the NDDEs can be intersected at a finite number of points, since this does not violate the uniqueness of the solutions for the DDE, an infintely-dementional system.   This further illustates the reason why the NDDEs represent these specific functions and other general continuous functions.
> ```
> Q2: Figure 2 could be more informative. As it stands, it only indicates that the initial state of a NDDE is specified by a function over an interval, rather than a single vector, which is all that is being used.
> ```
> **Response**: Many thanks for your advice. We have revised the Figure 2 to make it more understandable.
> ```
> Q3: Algorithm 1 is challenging to understand without some knowledge of how DDEs are solved.
> ```
> **Response**: Thanks for your comments. We added the details about the algorithm in the Appendix.
> ```
> Q4: In the experiments, NDDEs are only compared against NODEs, but it seems like ANODEs are the real competition. Notably in the image experiments performed in the ANODE paper they report substantially higher accuracies than NDDEs.
> ```
> **Response**: Many thanks for your careful reading and comments. As presented in “**Response to all reviewers**”, we conducted several additional experiments in the revised paper. In the very beginning, we are not going to compare against ANODEs. The reason is that in original paper of the ANODEs, the authors fixed the number of parameters of each model, but we think there are still many factors influencing the final performance of the experiments, such as the number of filters, number of the augmented dimensions, number of the parameters in the last layer and so on. Moreover, these factors may interact with each other. In our previous setting, the NODEs and NODEs are required the almost the same number of parameters in total along with the number of parameters in the last layer, which at least eliminates the influence of the last layer while the ANODEs with the augmented dimensions did not follow the same number of parameters in the last layer. For a fair comparison, we not only fix the total parameters almost the same but also the augmented dimension.  Overall, under the same augmented dimension, the rank of the performance for these models is A+NODE < A+NDDE < A+NODE+NDDE, revealing that the NDDEs may own the strong modeling capabilities and feature representations not only from the experimental aspect but also the theoretical aspect.

---

> > ### Author Response · Authors · 2020-11-21
> > **Thank you for your comments (2/2)**
> >
> > ```
> > Q5: This is mentioned in the discussion section, but I think the paper would benefit from an experiment involving a real-world dataset with some sort of delay component, ideally in which NDDEs outperform NODEs/ANODEs. The existing synthetic experiments are designed to show the strengths of NDDEs, while the image classification tasks don't substantially differentiate them from NODEs.
> > ```
> > **Response**: Many thanks for your comments. From the “**Response to all reviewers**”, we have proposed some extensions of the NDDEs, such as the NODE+NDDE, A+NDDE, A+NODE+NDDE, and provide a new theorem to support the modeling capabilities of the NDDEs. For the future research direction, we may apply our methods to the other real-world datasets properly.
> > ```
> > Q6:  The number of function evaluations is not reported anywhere, which seems like a significant statistic when NDDEs are substantially more complex than NODEs.
> > ```
> > **Response**: Thanks for your comments. We added the details of the number of function evaluations in the Appendix. The number of function evaluations for the NDDEs are much larger than the one of the ANODE.
> > ```
> > Q7:  Figure 7 could probably be improved by reducing the number of different parameter values reported (e.g. 3 columns instead of 6). The description also lists 5 parameter values but there are 6 columns.
> > ```
> > **Response**: Thanks for your careful reading and the helpful suggestion. We have modified the figure with a more accurate caption.
> > ```
> > Q8: The discussion paragraph "Extensions of the NDDEs" suggests a generalization to more complex delays, but all of the models in this paper ensure that the delayed value is always the constant initial value. Also, modeling the initial function as an ODE is mentioned, and I wonder whether the implication is that this would involve learning the initial function as a NODE itself. Either way, it's not obvious how either of these would yield better representations.
> > ```
> > **Response**: Great thanks for your comments.  From the “**Response to all reviewers**”, we proposed the extensions of NDDEs in the revised paper, including the NODE+NDDE, A+NDDE, A+NODE+NDDE. All these new models achieve the good performance on the experiments.  In addition, the idea of modeling the initial function of the NDDE as a NODE is that the constant function is special case of the NODEs, i.e., if the vector field of the NODE is zero. From such a viewpoint, modeling the initial function of the NDDE as a NODE may lead the dynamics of the NDDEs more complex, yielding better representations.  This better performacne is validated by the new conducted experiments shown in Figure 8 and Table 1.
> >
> > We would like to thank the reviewer again for his/her comments. We are confident that the quality of the revised paper has been sufficiently improved thanks to the reviewers’ comments and suggestions. Hopefully, the response and the revised paper have addressed the reviewer’s main concerns, and then reconsider the assessment in support of the revised paper. Finally, we will further answer any questions you may have.
> >
> > [1] . Emilien Dupont, Arnaud Doucet, and Yee Whye Teh. Augmented neural odes. In Advances in Neural Information Processing Systems, pp. 3140–3150, 2019.

---

### Official Review · AnonReviewer3 · 2020-10-31
**Using delay differential equations to improve the modeling capabilities of Neural ODEs**

**Rating:** 7
**Confidence:** 4

**Review:**

The paper builds on the Neural ODE framework by using delay differential equations (DDEs) instead of ordinary differential equations (ODEs). This can help in the modeling of systems with a time delay effect, and overcome many limitations of the Neural ODE framework.

Using DDEs is a novel technique in machine learning and can complement and build on the framework of Neural ODEs and help model systems with time delay dependencies and overcome some limitations of ODEs.

The paper was well written with a clear description of the model and theory to support it as well as the algorithm to train it. The experiments are well described though they lacked the model parameters and more description on their significance.

Major points:

1) Model parameters for the experiments were not listed. This can be especially important for systems with a time delay effect as in Figure 6 and 7. Was the same delay used as in the equation of each system? What happens when the delay is different?

2) While some complex systems were used to demonstrate the potential of the proposed model, as in Figure 7, the abstract claims that the model can successfully model chaotic dynamics yet that was not shown in the experiments. The dynamic regime for these systems with their chosen parameters were not clear together with the effect of time delay on these dynamics. Exploring this more closely can demonstrate the full potential of NDDEs.

3) The authors claim that unlike NODEs, NDDEs can overcome the problem of mutually intersected trajectories in phase space. They point to the experiment in Figure 6 as evidence of that. I feel there needs to be more discussion on this point both theoretically, proving that DDEs can indeed model intersections in phase space, and with a clearer demonstration experimentally.

---

> ### Author Response · Authors · 2020-11-21
> **Thank you for your comments**
>
> We would like to thank the reviewer for the overall positive feedback and the helpful suggestions.  We revised the paper carefully according to the reviewer’s comments. Our major changes in the revised paper are listed in “**Response to all reviewers**”.
> ```
> Q1: Model parameters for the experiments were not listed. This can be especially important for systems with a time delay effect as in Figure 6 and 7. Was the same delay used as in the equation of each system? What happens when the delay is different?
> ```
> **Response**: Thanks for the careful reading and the comments. The details of the model used in Figures 6 and 7 were added in the Appendix of the revised paper. Notably, for the synthetic datasets, we used the true delay of each system for the NDDEs. We think that how to accurately identify the intrinsic delay of each system or even jointly learn the underlying time-delay system might be of an extremely interesting topic.   Some model-free techinques in the literature for detecting unknown delays could be implemented.  In our settings, the main goal is to illustrate the difference between the NODEs and the NDDEs.  Particularly, we pointed out that the NODEs may not work for modeling the underlying time delay system.  Moreover, we conduct the additional experiments on the population dynamics and the Mackey-glass systems by using the ANODE method [Dupont2019].  As can be seen in the revised paper, its performance is better than that of a paricular NODE but still not that good for the long-range predictions.   This suggests that, because of the delay effect existing in the underlying system, the NODEs and ANODEs may not be suitable for such modelling.  That is, they are not likely to model the intrinsic dynamics of the underlying systems. We do believe that the NDDEs have the potential applications and need to be improved from the theoretical aspect and the efficiency of the algorithm.
> ```
> Q2: While some complex systems were used to demonstrate the potential of the proposed model, as in Figure 7, the abstract claims that the model can successfully model chaotic dynamics yet that was not shown in the experiments. The dynamic regime for these systems with their chosen parameters were not clear together with the effect of time delay on these dynamics. Exploring this more closely can demonstrate the full potential of NDDEs.
> ```
> **Response**: Thanks for your comments.  In the revision, we provided more examples including the chaotic models with particularly selected parameters and delays. Due to the limitation of the article length, refer to Section 1.4 and Section 3.3.1 in the book [1] for the bifurcation diagram for the MG system.
> ```
> Q3: The authors claim that unlike NODEs, NDDEs can overcome the problem of mutually intersected trajectories in phase space. They point to the experiment in Figure 6 as evidence of that. I feel that there needs to be more discussion on this point both theoretically, proving that DDEs can indeed model intersections in phase space, and with a clearer demonstration experimentally.
> ```
> **Response**Response: Many thanks for your insightful suggestion. We thereby added a theoretical result to prove that the NDDEs are universal, which may help the reviewer to understand the advantage of the NDDEs. It implies that the NDDEs can theoretically model some class of functions while NODEs cannot model them, as shown in Figure 4 and 6.
>
> Finally, we would like to thank the reviewer again for his/her time and positive feedback on the paper. The revised paper is improved a lot from the theoretical aspect and experimental aspect. We hope that the reviewer will be satisfied with the revised paper and the responses as well, and then consider revising the assessment in support of the revised paper. We may make further improvements according to your feedback.
>
> [1]. Thomas Erneux. Applied delay differential equations, volume 3. Springer Science & Business Media, 2009.

---

### Author Response · Authors · 2020-11-21
**Response to all reviewers**

We would like to thank the reviewers for your time, efforts, and valuable comments and suggestions, which do help us to significantly improve the quality of this work in several directions.  Accordingly, we try our best to make substantial revsions. The revised article now contains the analytical results, the extensions of the NDDEs, and the additional experiments on comparison studies.  More specifically, our main changes in the revision include:
1) We supplement an analytical result, presented in a form of Theorem, showing that the NDDEs are of universal approximators, which provides a convincing support for the strong representation capacity of the NDDEs.
2) We further provide several extensions based on the framework of the NDDEs, involving
   -  NODE+NDDE: modeling the initial function of the NDDEs as the ODEs,
   -  A+NDDE: augmenting the original space of the NDDE,
   -  A+ NODE+NDDE: augmenting the original space of the NODE+NDDE.

 Here, using the augmented models is inspired by the framework of the ANODE [Dupont2019]. We thereby compare these models on different tasks under the same dimensions for the augmented spaces and using the almost same model parameters.
3) We further conducted several additional experiments, including the new image dataset, SVHN. We mainly run the experiments on three kind of models, i.e., A+NODE (equivalently, ANODE), A+NDDE, A+NODE+NDDE, on the image datasets (CIFAR10, MNIST, SVHN) by changing the augmented dimensions.  In general, under the same augmented dimension and the almost same total model parameters, the performances for these models have the order: A+NODE < A+NDDE < A+NODE+NDDE. This clearly reveals that the NDDEs may have a stronger modeling capability and the feature representations. The detailed results can be found in the revision.  Also, it is worthwhile to mention that the performance of the ANODEs proposed in the paper is much lower than the one in the original paper [Dupont2019], due to the different choice of the augmented dimension and the number of the filters in the convolutional block of these models.
4) We revised the paper carefully and supplement more details for the experiments in the Appendix, including the model parameter/structures and the training configurations.
5) We almost modified all the figures and the tabular in the main text to provide the much clearer plots and make them more informative as well.
6) We provided more discussions on the NDDEs and showed the future directions in the revision.

Finally, we thank alll the reviewers again for your insightful comments. We do believe that the revised article is much improved not only from the theoretical aspect but also from the experimental aspect.  In addition, we hope that the revised article as well as the individual responses for each reviewer adequately addresses the reviewers’ concerns. We appreciate it very much if the reviewers could find the further contributions and our efforts on this revised work.

---

### Decision · Program_Chairs · 2021-01-07
**Final Decision**

**Decision:**

Accept (Poster)

**Comment:**

This paper is a variant of the large growing class of Neural ODEs, and adds dependency on a time delay to the baseline, which allows to model a larger class of physical systems, in particular adding the possibility of crossing paths in phase space.

After initial evaluation, the paper was on the fence, with 2 reviewers providing favorable reviews, and 2 reviewers recommending rejection. A particular important issue raised was positioning with respect to prior art, [Dupont 2019], with some substantial overlap between the papers; requests of theoretical discussions of the class of studied systems and its properties.

Most of these remarks have been addressed by the authors, in particular positioning and experimental comparisons.

The AC judged that the paper had been sufficiently improved and recommends acceptance.